# Nucleosome spacing can fine-tune higher-order chromatin assembly

Lifeng Chen [1,2], M. Julia Maristany [2,3,4,9], Stephen E. Farr[3,9], Jinyue Luo[5], Bryan A. Gibson[1,2,6], Lynda K. Doolittle[1], Jorge R. Espinosa [2,4,7], Jan Huertas [2,4,8], Sy Redding [2,5], Rosana Collepardo-Guevara [2,3,4,8] ✉ & Michael K. Rosen [1,2] ✉

Cellular chromatin displays heterogeneous structure and dynamics, properties that control diverse nuclear processes. Models invoke phase separation of conformational ensembles of chromatin fibers as a mechanism regulating chromatin organization in vivo. Here we combine biochemistry and molecular dynamics simulations to examine, at single base-pair resolution, how nucleosome spacing controls chromatin phase separation. We show that as DNA linkers extend from 25 bp to 30 bp, as exemplars of 10 N + 5 and 10 N (integer N) bp lengths, chromatin condensates become less thermodynamically stable and nucleosome mobility increases. Simulations reveal that this is due to trade-offs between inter- and intramolecular nucleosome stacking, favored by rigid 10 N + 5 and 10 N bp linkers, respectively. A remodeler can induce or inhibit phase separation by moving nucleosomes, changing the balance between intra- and intermolecular stacking. The intrinsic phase separation capacity of chromatin enables fine tuning of compaction and dynamics, likely contributing to heterogeneous chromatin organization in vivo.

The fundamental organizing unit of the mammalian genome is the nucleosome, composed of ~147 base pairs of DNA wrapped around a histone octamer[1–3]. Nucleosomes are joined together by linker DNA to form chromatin fibers, which condense successively to form higher-order structures and functional domains inside the nucleus[3–5]. Despite substantial compaction, nucleosomes remain dynamic in cells, exhibiting functionally important local diffusion at short length scales[6–9]. The structural and physical mechanisms that produce compaction while retaining local dynamics are incompletely understood.

Fluorescence and electron micrographs have revealed that in most cells chromatin is heterogeneously folded without extensive long-range order (e.g., 30 nm fibers)[10–14]. Irregular domains of high nucleosome density are connected by areas of sparse nucleosomes[6,10–13,15,16]. Higher resolution cryo-electron micrographs of cell nuclei and purified native chromatin have revealed positions and orientations of condensed nucleosomes and have enabled the tracing of long chromatin fibers[11–13]. These data showed that nucleosome-nucleosome orientations are irregular, with some clusters of a few nucleosomes (2-4) engaging in face-to-face stacks within a fiber or between fibers. Additionally, linker DNA length is variable and has been proposed to contribute to the observed structural heterogeneity[17–20]. Mirroring the structural heterogeneity, nucleosome dynamics are also heterogeneous, with different rates of movement observed in different locations, and dependent on the chromatin state, such as histone acetylation, and active transcription[21,22]. The

[1]Department of Biophysics and Howard Hughes Medical Institute, University of Texas Southwestern Medical Center, Dallas, TX, USA. [2]Marine Biological Laboratory Chromatin Collaborative, Marine Biological Laboratory, Woods Hole, MA, USA. [3]Maxwell Centre, Cavendish Laboratory, Department of Physics, University of Cambridge, Cambridge, UK. [4]Yusuf Hamied Department of Chemistry, University of Cambridge, Cambridge, UK. [5]Department of Biochemistry and Molecular Biotechnology, University of Massachusetts Chan Medical School, Worcester, MA, USA. [6]Department of Cell and Molecular Biology, St. Jude Children's Research Hospital, 262 Danny Thomas Place, Memphis, TN, USA. [7]Department of Physical Chemistry, Complutense University of Madrid, Madrid, Spain. [8]Department of Genetics, University of Cambridge, Cambridge, UK. [9]These authors contributed equally: M. Julia Maristany, Stephen E. Farr. ✉e-mail: rc597@cam.ac.uk; michael.rosen@utsouthwestern.edu

factors that control compact versus sparse nucleosome organization and differences in dynamics are not fully understood.

Native chromatin is highly complex, with location-specific variations in histone types, nucleosome positions, DNA sequences, and epigenetic marks[23–26]. These variations complicate the understanding of the physical drivers of chromatin behaviors. Simplified in vitro systems using reconstituted nucleosome arrays provide opportunities to isolate specific features of the chromatin fiber and understand their effects on chromatin behavior[27–33]. Studies of such systems have revealed the intrinsic capacity of chromatin to undergo salt-dependent oligomerization reversibly and cooperatively[29,31,34–37]. This oligomerization leads to liquid-liquid phase separation (LLPS) in physiologic conditions; the resulting droplets recapitulate many aspects of cellular chromatin[38,39]. These include compaction that is reversible, cooperative, and dynamic, and which responds to regulatory histone tail modifications and chromatin-binding proteins. Studies of these systems also revealed a periodic relationship between the length of internucleosomal linker DNA and LLPS propensity[38].

Eukaryotic genomes demonstrate an oscillatory pattern of linker length frequencies, with an enrichment for 10 N + 5 (integer N, e.g., 15, 25, etc.) linkers and a depletion of 10 N (e.g., 10, 20) linkers[40–43]. For DNA linkers shorter than ~40 base pairs, the rigid nature of the DNA double helix torsionally constrains successive nucleosomes. Since B-form DNA has 10.4 base pairs per helical turn, nucleosomes connected by a 10 N linker are separated by approximately an integral number of turns. The resulting orientation allows favorable face-to-face stacking of alternating nucleosomes in an array, producing the canonical 30 nm fiber structure[17,18,44–47]. In contrast, nucleosomes connected by a 10 N + 5 linker are rotated 180° from each other, producing a configuration that does not form regular higher-order structures resolvable by crystallography or electron microscopy[45]. Studies of arrays with linkers spanning 10 N and 10 N + 5 have revealed a periodic relationship, where linker lengths approaching 10 N afford more compact, regular, and energetically stable fibers[17,48,49]. Linker length also modulates LLPS propensity, as 10 N + 5 arrays phase separate more readily than 10 N arrays[38]. It remains unexplored how fine differences between the two extremes affect the structural and thermodynamic features of individual fibers, and consequently chromatin condensation and dynamics.

Here we used biochemistry and computer simulations to examine, at single base-pair resolution, how internucleosomal linker length affects LLPS of reconstituted chromatin arrays (Fig. 1a). As linker length increases from 25 to 30 bp, the LLPS threshold salt concentration and dynamics within the condensates increase, in non-linear fashion. Multiscale simulations reproduced both trends. In the simulations, the 25 bp arrays inside condensates readily form energetically favorable face-to-face stacking interactions between individual fibers (i.e., inter-fiber). In contrast, the 30 bp arrays make face-to-face interactions mostly within individual fibers (i.e., intra-fiber) and make less favorable face-to-side and side-to-side interactions between fibers. The relative proportions of these interaction modes shift, along with conformational preferences of individual arrays, in non-linear fashion as linkers increase from 25 bp to 30 bp. These differences rationalize the phase separation propensities and dynamics as arising from a balance between intra- and inter-fiber nucleosome stacking interactions. Finally, we demonstrate experimentally and computationally that the *Drosophila* ISWI nucleosome remodeler can induce or inhibit phase separation of chromatin by moving nucleosomes to favorable or unfavorable spacing, respectively. Our study highlights internucleosomal linker length as a dynamically regulatable epigenetic parameter that, with single base-pair precision, dictates the conformation of the chromatin fiber and, consequently, its higher-order organization and dynamics. Variation in linker length could thus contribute to heterogeneity of chromatin organization in the nucleus.

## Results

### Simulations recapitulate the LLPS threshold differences between 10 N and 10 N + 5 arrays

We used a turbidity assay, which monitors scattering of 300 nm light by phase-separated droplets in solution, to quantify phase separation of reconstituted dodecameric nucleosome arrays based on the Widom 601 positioning sequence[28]. Initially, we examined arrays with linker lengths of 25 or 30 base pairs, as representatives of the 10 N + 5 and 10 N classes, respectively (Fig. 1b, for brevity, henceforth referred to as 25 bp or 30 bp arrays). In low salt buffer the arrays exhibit baseline scattering of 300 nm light. As salt is increased (KOAc or Mg(OAc)$_2$), condensate formation coincides with a sharp increase in scattering (observed as droplet formation by fluorescence microscopy, Supplementary Fig. 1). At 2 mM Mg$^{2+}$, the 25 bp array phase separates with a KOAc threshold concentration of 54 ± 1 mM (see Methods), but the 30 bp array does not phase separate up to 150 mM KOAc (at 42 nM array/500 nM nucleosome; Fig. 1b, Supplementary Data 2). We note that 30 bp arrays that are under-assembled, i.e., contain detectable histone hexamer (Supplementary Fig. 2a), do phase separate at low magnesium concentrations (Supplementary Fig. 2b), a behavior we have not further pursued. In the presence of 100 mM KOAc, well-assembled 30 bp arrays phase separate at considerably higher magnesium concentrations: the phase separation threshold for the 25 bp array is 0.1 ± 0.03 mM Mg$^{2+}$, while that of the 30 bp array is 2.7 ± 0.06 mM Mg$^{2+}$ (Fig. 1c, Supplementary Data 2).

We previously developed a multiscale model of chromatin that exploits the ability of atomistic representations to accurately describe the behavior of nucleosomes, histone proteins, and DNA, and the computational efficiency of simplified coarse-grained representations to predict mesoscale properties of chromatin arrays and emergent properties of chromatin solutions[50]. Our approach includes coarse-grained models at two resolutions. A near-atomistic *chemical-specific chromatin model* describes each amino acid and DNA base pair explicitly and captures the effects of both amino acid and DNA sequence variations on the structure of individual nucleosome arrays. A *minimal chromatin model* simplifies the description of nucleosomes (from ~1500 particles to ~30 particles) and can probe the phase behavior of chromatin solutions while maintaining physicochemical accuracy.

We applied the multiscale model to perform direct coexistence molecular dynamics simulations of solutions containing dodecameric 25 bp and 30 bp arrays (Fig. 1d). As in our previous study of an 18 bp array[38], we found that both the 25 bp and 30 bp arrays form homogeneous low-density solutions at low salt concentrations, but separate into dense and dilute coexisting phases above a threshold salt concentration. By repeating the simulations at multiple salt concentrations, we mapped the coexistence curves for the two systems in the plane of salt concentration versus chromatin density (Fig. 1d). The minimum of the coexistence curve represents the critical salt and chromatin concentrations. These phase diagrams recapitulate the experimental results: the 25 bp arrays have lower critical salt concentrations than the 30 bp arrays (Fig. 1d). A lower critical salt concentration indicates that the chromatin condensates are stable under a larger fraction of the parameter space (Fig. 1d; shaded areas); i.e., they exhibit higher thermodynamic stability.

In simulations involving additional linker lengths the model accurately predicts the experimental observations that 10 N + 5 arrays phase separate more readily (at lower salt) than 10 N arrays (Fig. 1d). Thus, both experiments and simulations show that phase separation is favored by linker lengths that produce a residual half-turn of DNA between successive nucleosomes.

### Single base-pair mapping of the phase separation threshold and droplet dynamics

Given the relationship between linker length and the helical pitch of DNA, we anticipated that the 25 bp and 30 bp systems would reflect

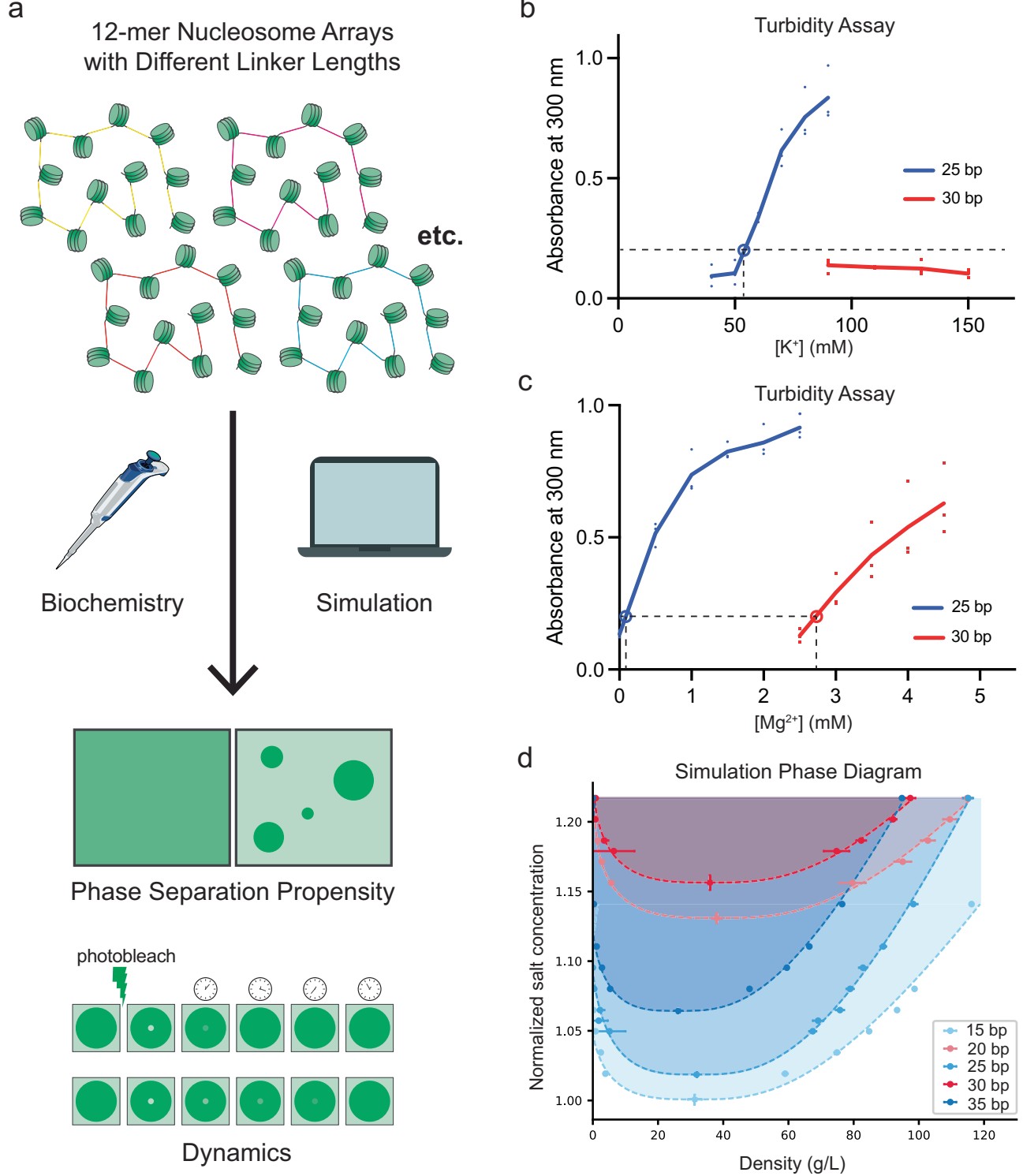

**Fig. 1 | Simulations recapitulate the LLPS threshold differences between 10 N and 10 N + 5 arrays. a** Design of this study. Nucleosome arrays of various linker lengths are investigated in vitro and in silico to determine their phase separation propensity and dynamics. **b** Turbidity assay for 25 bp and 30 bp arrays. Solid data points represent the absorbance measured at 300 nm for arrays in 2 mM Mg$^{2+}$ and K$^+$ concentration indicated. The open data point represents the interpolated threshold concentration at which absorbance equals 0.2. $n = 3$ independent experiments. **c** As in **b** except arrays are in 100 mM K$^+$ and Mg$^{2+}$ concentration indicated. In **b** and **c** dashed lines indicate phase separation thresholds inferred from the data. **d** Simulated phase diagrams (binodals, salt concentration versus chromatin density) for chromatin solutions with 15 bp, 20 bp, 25 bp, 30 bp, or 35 bp

linker lengths. Circular points represent chromatin density in dilute phase (left branch) and dense phase (right branch) at a given monovalent salt concentration. Each binodal is normalized based on the critical salt concentration of the 15 bp chromatin solution (65 mM NaCl). Density is defined as the molar mass of chromatin molecules per unit of volume in g/L. $n = 10,000$ recordings from 100 million timesteps. Data are represented as mean ± s.d. (error bars are smaller than the symbols in most cases). Critical points are calculated by fitting the data to Eqs. (7) and (8), and the error bars are ± error from the least-squares fitting procedure. The dashed lines represent coexistence curves and are obtained by fitting the data to Eqs. (7) and (8). The shaded area indicates the region of two-phase coexistence, where chromatin phase separation is thermodynamically stable.

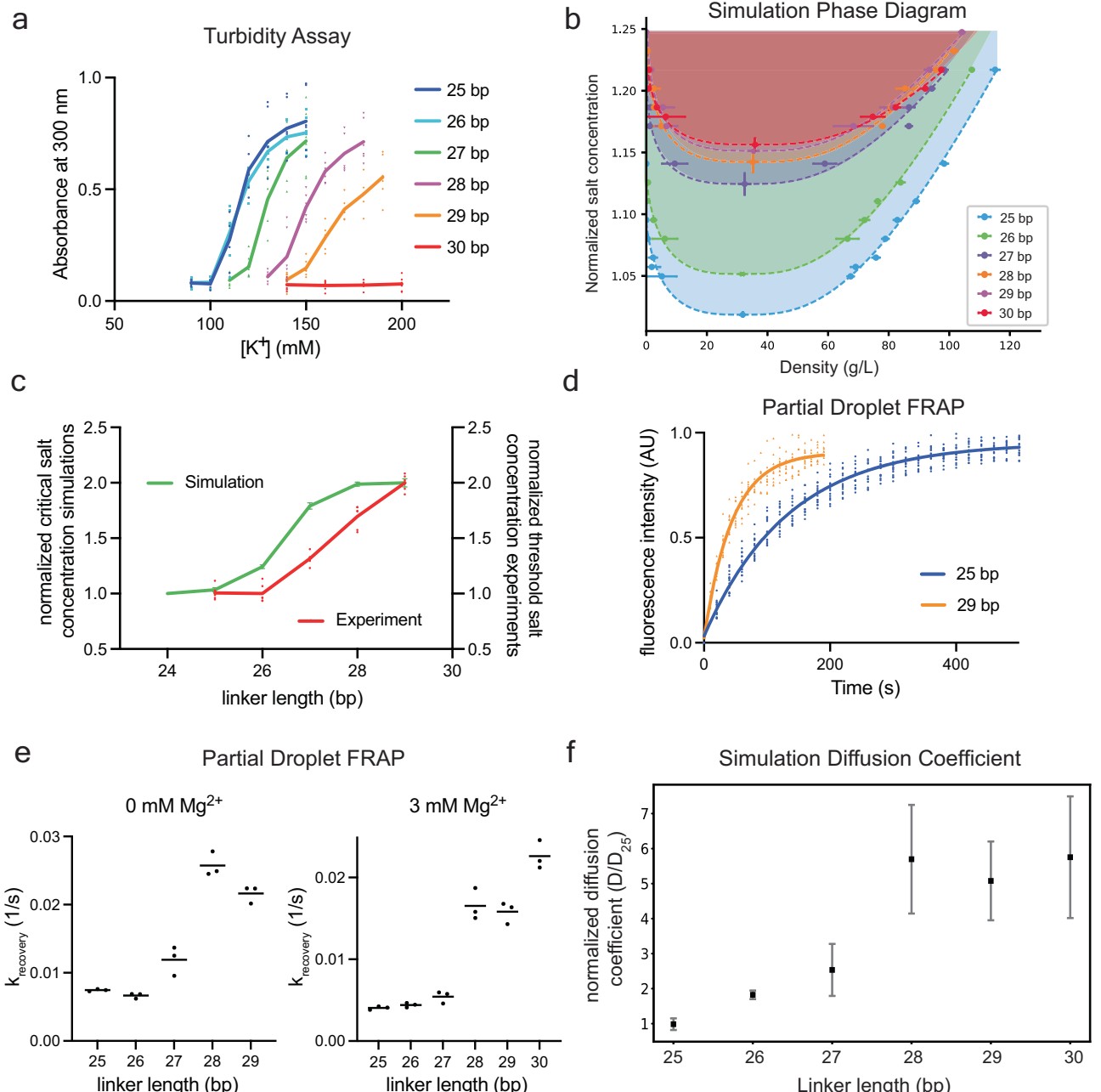

**Fig. 2 | Simulations recapitulate the fine details of 25–30 array LLPS. a** Turbidity assay for arrays with linker lengths from 25 – 30 bp. Each data point represents the absorbance measured at 300 nm for arrays in 0 mM $Mg^{2+}$ and $K^+$ concentration indicated. $n = 6$ independent experiments. **b** Simulated phase diagrams (salt concentration versus chromatin density) for chromatin solutions with 25 – 30 bp linker DNA lengths. Phase diagrams were computed and are represented as in Fig. 1e. $n = 10,000$ recordings from 100 million timesteps. Data are represented as mean ± s.d. **c** Red points show experimental $K^+$ interpolated thresholds for chromatin arrays of indicated linker lengths from turbidity assays in **a** normalized from 1 to 2. $n = 6$ independent experiments. Green points show simulated critical monovalent salt concentrations for chromatin arrays of indicated linker lengths, normalized from 1 to 2, and are represented as normalized mean ± s.d. $n = 10,000$ recordings from 100 million timesteps. **d** Fluorescence recovery over time of a central

bleached region in condensates composed of arrays of 25 bp or 29 bp linkers. Data are represented as normalized mean fluorescence intensity for 15 droplets (25 bp) or 10 droplets (29 bp). **e** Rate constant of fluorescence recovery for chromatin condensates of indicated linker lengths. Rate constant is calculated by fitting the fluorescence intensity vs time to a single exponential. $n = 3$ independent experiments. **f** Simulation-derived diffusion coefficients for chromatin arrays inside the condensed phase as a function of linker DNA length at 1.23 salt concentration, computed using the minimal chromatin coarse-grained model by measuring the mean square displacement (MSD) from simulations of the pure condensed phase at the coexistence density. Diffusion coefficients are obtained from the slope of the MSD line, normalized to the value of 25 bp. Data are represented as mean ± s.d. Error bars were computed using block averaging over the entire simulation trajectory, consisting of $n = 700$ points.

two extremes, defined by opposite orientations of successive nucleosomes. To understand how phase separation behaviors vary between these extremes, we generated a series of arrays with single base-pair steps in linker length, from 26 to 29 bp. As above, we used a turbidity assay to determine the LLPS threshold concentration of

KOAc for each array (in the absence of $Mg^{2+}$). As shown in Fig. 2a, the LLPS threshold is very similar for the 25 bp and 26 bp arrays, but increases progressively from 26 bp to 29 bp (the 30 bp array does not phase separate at these solution conditions and concentrations without $Mg^{2+}$). A similar pattern is observed when titrating $Mg^{2+}$ at

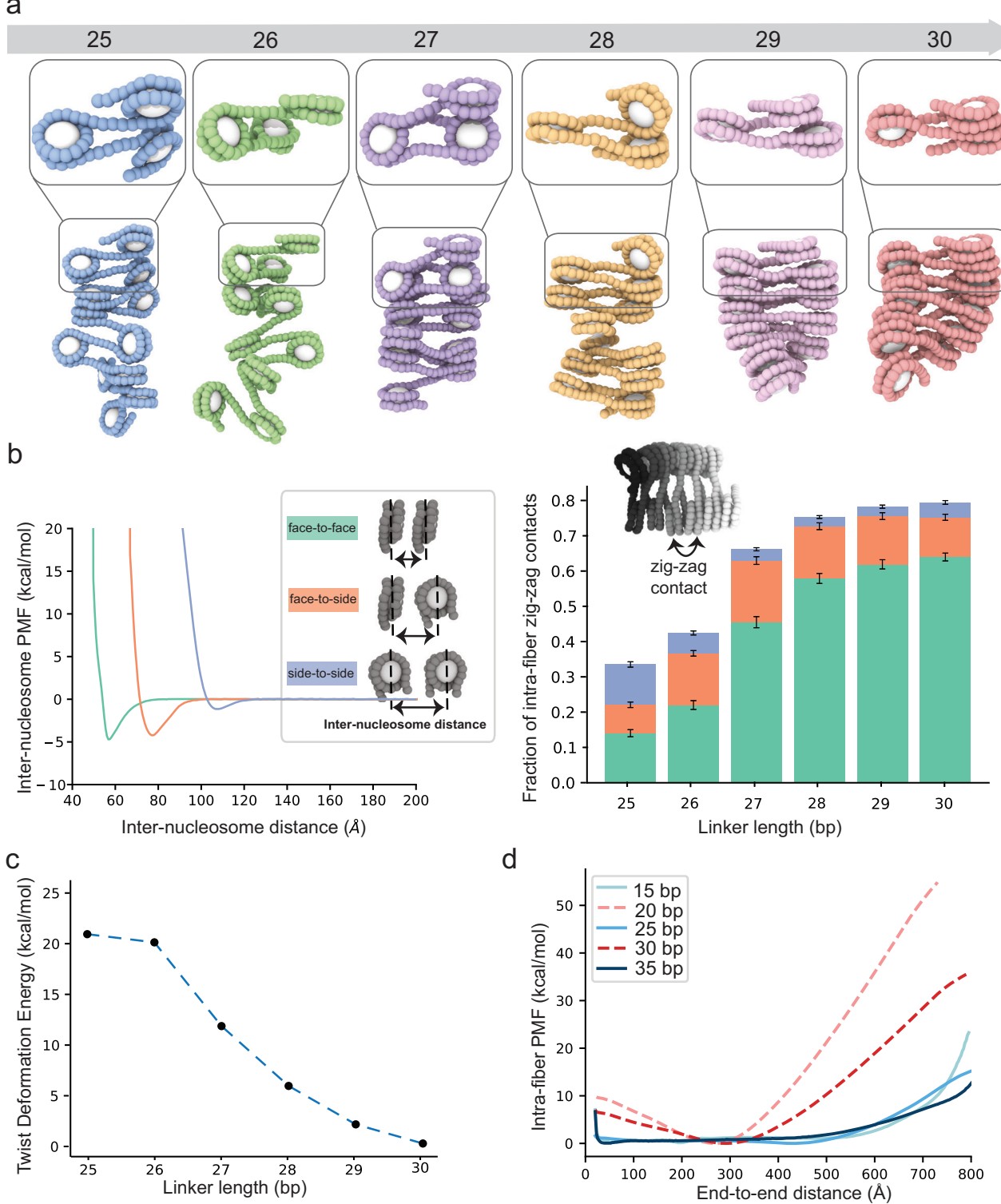

100 mM KOAc, where the 30 bp array also phase separates (Supplementary Fig. 3).

We used our multiscale model to construct phase diagrams for each of these arrays in the presence of monovalent salt (the modeling cannot currently account for divalent cations). These simulations showed good qualitative agreement between the predicted critical salt concentration of the phase diagram and the experimental KOAc phase separation threshold concentration (Fig. 2a–c). Both show a non-linear increase in threshold salt as linker DNA increases. The modeled

threshold is systematically shifted by -1 bp compared to the observed values, likely due to the simplified representation of solvent and ions (see Methods), but the overall trend clearly mirrors the experimental data closely (Fig. 2c).

We also used fluorescence recovery after photobleaching (FRAP) to learn how the dynamics of chromatin droplets are affected by linker length. As shown in Fig. 2d, Supplementary Fig. 4–5, fitting the recovery data to single exponentials reveals that, in the absence of Mg²⁺, the recovery rate constant increases (i.e., dynamics increase)

**Fig. 3 | Internucleosomal linker length dictates single fiber structures.**
**a** Representative simulation structures (bottom) and tri-nucleosome elements (top) of 12-nucleosome chromatin arrays inside the condensed phase from Direct Coexistence simulations at 1.23 normalized salt concentration. Histone core ellipsoids are shown in white and DNA beads belonging to chromatin arrays with varying linker DNA lengths are colored: 25 bp (blue), 26 bp (green), 27 bp (purple), 28 bp (orange), 29 bp (pink), and 30 bp (red). **b Left** Potential of mean force (PMF) computed between nucleosome pairs at three orientations and at 150 mM salt, using umbrella sampling simulations of the chemical-specific coarse-grained model. Inset illustrates the three types of pairwise nucleosome interactions considered: face-to-face (green), face-to-side (orange), and side-to-side (blue). **Right** fraction of intra-fiber contacts among second-nearest linear nucleosome neighbors (zig-zag) as a function of linker DNA length, computed from the simulations in panel **a**. Two nucleosomes are considered to be "in contact" if they belong to the same fiber and their geometric centers are closer than 120 Å. The fraction represents the total number of zig-zag

contacts along the simulation trajectory divided by the total number of nucleosomes in one fiber and the total number of frames per trajectory. The fraction of contacts is divided according to the contact type as described in Methods. $n = 120$ independent chromatin fibers. Data are represented as mean ± s.d. **c** Energy cost of twisting the linker DNA to enforce a parallel orientation among sequential nucleosomes and enable face-to-face stacking. This twist deformation energy reflects the additional energy required to override the natural twist angle between adjacent nucleosomes— i.e., that dictated by the length and twist of the linker DNA—and enable the formation of a perfect zig-zag contact. The twist deformation energy is calculated using the rigid base-pair model, focusing on twist-twist stiffness and accounting for the deviation of the total helical twist of the linker DNA from an integral number of turns. **d** PMFs as a function of the end-to-end extension of single chromatin arrays with different linker lengths at 150 mM monovalent salt using umbrella sampling simulations of the minimal chromatin coarse-grained model. The end-to-end extension is the distance between the first and last DNA bead.

~3-fold as the linker grows from 25 bp to 29 bp (Fig. 2e, Supplementary Fig. 4, Supplementary Data 2). Similar to the phase separation threshold, the change in dynamics is non-linear, with a relatively sharp decrease in recovery time between 26 and 28 bp. In the presence of 3 mM Mg$^{2+}$, the change in recovery rate constant is larger, ~5-fold, and sharper, essentially dividing the arrays into a slow-recovery group, 25 bp – 27 bp, and a fast-recovery group, 28 bp – 30 bp (Fig. 2e, Supplementary Fig. 5, Supplementary Data 2).

Similar dynamic behaviors were observed in the simulations. We assessed dynamics by computing diffusion coefficients of individual arrays within the condensates from the mean square displacement of their molecular center of mass (Fig. 2f). We also performed additional simulations of the isolated condensates (without surrounding bulk) at array densities observed for the dense phase in the coexistence simulations. In excellent agreement with experiment, the simulated diffusion coefficients increase sharply from 26 to 28 bp and follow a sigmoidal shape (Fig. 2f). The 30 bp chromatin has consistently higher dynamics than 25 bp chromatin at multiple salt concentrations (Supplementary Fig. 6).

In summary, the higher-order assembly of chromatin varies with linker length, showing changes in phase separation threshold and droplet dynamics with single base-pair increments. The changes in dynamics can be sharp, shifting between two classes in a single base pair (27-28 bp). Simulations capture the experimental results, showing decreasing phase separation propensity and faster array diffusion inside condensates as the linker length increases from 25 bp to 30 bp.

## Simulations suggest structural and energetic explanations for biochemical behaviors

The simulations can provide structural and energetic insights into the mechanisms driving chromatin phase separation. Double helical DNA is torsionally rigid on length scales of the linkers examined here (8-10 nm), imposing substantial rotational constraint on the orientations of adjacent nucleosomes. Prior studies have shown that the orientations imparted by successive, equal-length 10 N linkers create zig-zag stacking of every other nucleosome within an array, producing a two-start 30 nm fiber structure[17,44–47]. Snapshots of single chromatin fibers taken from the simulations of the 30 bp arrays also show this two-start structure (Fig. 3a, red; Supplementary Fig. 7). In contrast, prior studies have demonstrated that 10 N + 5 arrays are unevenly folded in a manner that prevents intra-fiber stacking[17]. This again is observed in snapshots from our simulations of the 25 bp arrays, which show heterogeneous conformations with little zig-zag stacking (Fig. 3a, blue, Supplementary Fig. 7). Our recent cryo-electron tomographic analyses of chromatin condensates further confirm these conformational differences between the 25 bp and 30 bp arrays[51]. The additional snapshots in Fig. 3a reveal intermediate behaviors between the two extremes: the 26 bp arrays (green) resemble the more irregular folding of the 25 bp arrays, the 28 bp (orange) and 29 bp (violet) arrays present

two-start structures, and the 27 bp arrays (purple) combine zig-zag dominant stacking with some irregularity.

We quantified the diversity of structures by determining the prevalence of three stereotypical types of nucleosome-nucleosome interactions: face-to-face, face-to-side, and side-to-side (Fig. 3b, left). We find that 28 – 30 bp arrays engage in substantially more face-to-face intra-fiber interactions compared to 25 – 27 bp arrays (Fig. 3b, right). To rationalize the origin of this difference, we characterized fiber energetics in two different ways. First, we computed the potential of mean force (PMF) as a function of inter-nucleosome pairwise distance for the three interaction geometries. We found that the PMF is minimal (providing greatest thermodynamic gain) for face-to-face interactions (−4.8 kcal/mol), followed closely by face-to-side (−4.2 kcal/mol), and more distantly by side-to-side (−0.8 kcal/mol), with optimal interaction distances of 60 Å, 75 Å and 110 Å, respectively (Fig. 3b, left). Second, as the linker length shortens from 30 to 25 bp, the orientations imposed by the linker DNA between sequential nucleosomes shift by ~36° per base pair—from 0° for parallel (30 bp), which promotes face-to-face stacking, to 180° for anti-parallel (25 bp), which hinders it. To quantify this effect, we calculated the energy cost of twisting the linker DNA to enforce a parallel face-to-face orientation of nucleosomes as a function of linker length (Fig. 3c). Our results show a non-linear decrease in the energy required to overcome the DNA torsional rigidity with increasing linker length: the cost is four times higher than the strength of face-to-face stacking for 25 and 26 bp linkers (21 and 20 kcal/mol, respectively), more than double for 27 bp (12 kcal/mol), and progressively lower for 28, 29, and 30 bp (6, 2, and 0.3 kcal/mol). Combining these two energetics, the dominant stacking behavior in 28 – 30 bp arrays arises because the enthalpic gain from forming face-to-face nucleosome interactions is sufficient to overcome the enthalpic penalty introduced by twisting these DNA linkers. In contrast, for 25/26 bp arrays, the energy cost of deforming the DNA to orient nucleosomes parallel to each other becomes too large, preventing stable stacking. Consistently, the enthalpic gain from face-to-face stacking for 10 N arrays manifests in appreciably greater energetic cost of extension in single-molecule force-extension simulations (Fig. 3d, Supplementary Fig. 8), concordant with previously reported single-molecule force spectroscopy measurements[48].

These differences in the folding of single fibers then produce differences in inter-fiber interactions within the condensed phase observed in the coexistence simulations. For the 30 bp arrays, since most nucleosome faces are sequestered intramolecularly (Fig. 3a, b), inter-fiber contacts involve mostly nucleosome edges of relatively ordered molecules (Fig. 4a, b). In contrast, since intra-fiber face-to-face nucleosome stacking is disfavored within the 25 bp arrays (Fig. 3a, b), nucleosome faces are free to interact with neighboring molecules, producing heterogeneous collections of face-to-face and face-to-side contacts (Fig. 4a, b). To further quantify these observations, we measured the binding free energy between pairs of fibers by coupling the umbrella

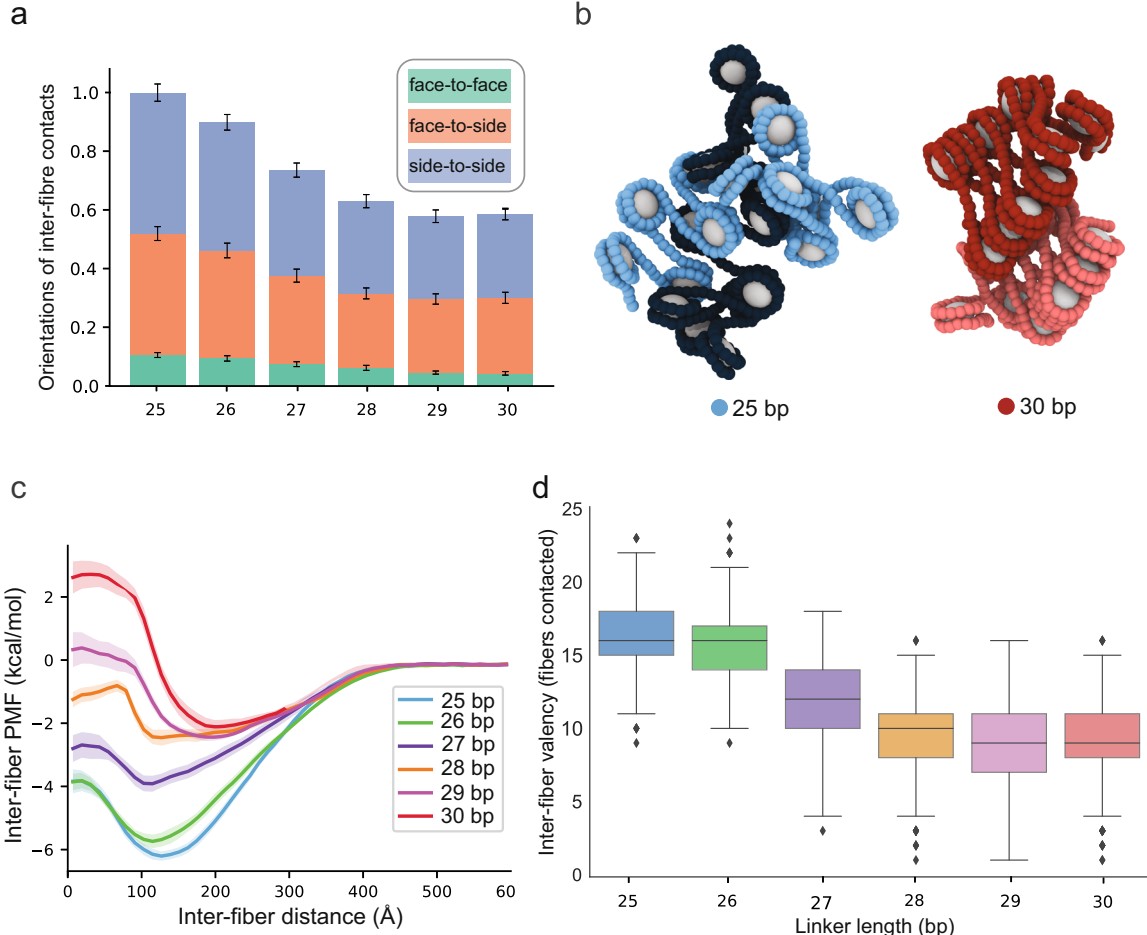

**Fig. 4 | Internucleosomal linker length dictates higher-order chromatin fiber structure and energetics. a** Fraction of inter-fiber contacts as a function of linker length computed from the simulations in panel 3a. Two nucleosomes are considered to be "in contact" if they belong to different fibers and their geometric centers are closer than 120 Å. The fraction represents the total number of contacts along the simulation trajectory normalized by the maximum value for the 25 bp system, in order to obtain a value ranging from 0 to 1. The fraction of contacts is divided according to the contact type: face-to-face (green), face-to-side (orange), and side-to-side (blue). $n = 10,000$ recordings from 100 million timesteps. Data are represented as mean ± s.d. **b** Representative snapshots of the interaction at the PMF minima for the 25 bp and 30 bp chromatin. **c** PMFs describing the change in interaction free energy for pairs of chromatin arrays with linker lengths 25 – 30 bp as a function of the pairwise inter-fiber distance. PMFs were computed from umbrella sampling simulations coupled with temperature replica exchange molecular dynamics at 150 mM monovalent salt. $n = 5$ independent repeats. Solid curves represent the mean, and the shading represents the standard deviation. **d** Inter-fiber valence as a function of linker DNA length computed from the simulations described in panel 3a. Valence is defined as the average number of neighboring chromatin arrays that each fiber contacts inside the condensed phase. Box plots show, for each linker length, the median value as well as the distribution of the data by indicating the upper and lower quartile values, and the maximum and minimum values. Outliers are represented with dots. $n = 120$ independent chromatin fibers.

sampling method with temperature replica exchange molecular dynamics simulations. The thermodynamic gain from inter-fiber interactions is greater for 25 bp arrays than for 30 bp arrays (Fig. 4c, Supplementary Fig. 9), consistent with the greater capacity of the former to make the energetically more favorable face-to-face and face-to-side contacts with neighbors (Fig. 4a). In contrast, the 30 bp fibers are "self-satisfied", leaving mostly the weaker side-to-side contacts between fibers (Fig. 4a). Gradually increasing the linker DNA length from 25 to 30 bp favors intra-fiber face-to-face contacts in a non-linear fashion (Fig. 3b), thereby reducing the free energy of binding among chromatin fibers in a similar non-linear fashion (Fig. 4c, Supplementary Fig. 9): the 25 and 26 bp arrays establish the strongest interactions (~−6 kcal/mol), the 27 bp arrays form moderately strong contacts (~−4 kcal/mol), and the 28 – 30 bp arrays bind to each other weakly (~−2 kcal/mol). Furthermore, energetic differences between pairs of fibers are magnified by the valency of fibers in the condensate, where interactions between 25 bp arrays are not only more favorable, but also more numerous (Fig. 4d).

Together, the simulations provide a model to explain the different behaviors of arrays with linkers ranging from 25 bp to 30 bp.

Interactions involving nucleosome faces are stronger than those only involving nucleosome sides. The conformation of the 30 bp array enables stacking of alternate nucleosomes, so the strongest inter-nucleosome contacts occur in intramolecular fashion. In contrast, the conformation of the 25 bp array frees nucleosome faces, enabling the strongest contacts to occur intermolecularly. Since phase separation is driven by intermolecular binding, the 25 bp arrays undergo LLPS more readily. Additionally, since internal dynamics are dictated by the rates of molecular dissociation, which are typically inversely related to interaction strength, these same features afford slower dynamics of the 25 bp array droplets. The non-linear shifting from intra-array to inter-array stacking as linkers are decreased from 30 bp to 25 bp explains the non-linear progression of behaviors across the series.

## Simulation of chromatin fibers with longer and heterogeneous linkers

We also used simulations to extrapolate the behaviors of arrays with longer and heterogeneous linkers. We first determined pairwise

inter-fiber interaction energies for linkers from 15 bp to 65 bp. The trade-offs between inter- and intramolecular face-to-face/side contacts observed in the 25 bp – 30 bp series persist across the wider span of linker lengths, producing an oscillatory pattern (Fig. 5a). This pattern is due to constraints imposed by DNA torsional rigidity (Fig. 5b), which enable only zig-zag contacts for certain linker lengths (Fig. 5c), since relaxing rigidity abolishes the oscillations by enabling formation of zig-zag contacts for all linker lengths (Fig. 5d). Both with and without torsional rigidity, the free energy of association increases (interactions become less favorable) as linker lengths increase, due to a combination of increased linker-linker repulsion and a decrease in the nucleosome-nucleosome interaction density along the fiber (Figs. 5a, 5d). The inter-fiber nucleosome interaction energy correlates well with the computed critical salt concentration for phase separation, with the same oscillatory pattern and overall decreasing thermodynamic drive as linker length increases (Fig. 5e). As for the inter-fiber interactions, the oscillatory pattern of the critical salt concentration is abolished when we compute phase diagrams based on chromatin with linker DNA lacking torsional rigidity (Fig. 5f).

To further test the principle that the tendency to form intramolecular stacks hinders intermolecular interactions and LLPS, we simulated two dodecameric nucleosome arrays with heterogeneous linker lengths patterned as $[L + 2, L − 2, L]_3[L + 2, L − 2]$, where L represents either 20 bp or 25 bp (henceforth referred to as 20 ± 2 bp and 25 ± 2 bp arrays, respectively). The average linker length of these arrays falls into the 10 N or 10 N + 5 class, respectively. However, the +2 and −2 bp deviations allow for stacking of N and N + 2 nucleosomes in both cases (Supplementary Fig. 10). Since the 20 ± 2 bp and 20 bp arrays both form intra-fiber stacks, the two array types have similar calculated PMF- and coexistence curves (Supplementary Fig. 10). In contrast, because the 25 ± 2 bp arrays form intra-fiber stacks but the 25 bp arrays do not, the former shows substantially weakened intermolecular interactions in PMF calculations and diminished capacity to phase separate compared to the latter (Supplementary Fig. 10). This result shows that a 2 bp deviation can have different impacts on different array types, and that the regular stacking structure of 10 N fibers can be less sensitive to certain perturbations than the irregular conformations of 10 N + 5 fibers. These simulations reinforce the idea that the balance between intra- and inter-fiber interactions dictates array-array interactions and LLPS propensity.

## Nucleosome remodeling can modulate LLPS

Given the differences in phase separation propensity of the different linker length variants, we asked whether a remodeling enzyme that translocates nucleosomes along a DNA strand could control condensate formation. We generated 25 bp and 30 bp arrays that contain 52 bp of free DNA beyond the terminal nucleosome (Supplementary Data 1). In mononucleosome remodeling assays, the *Drosophila* ATP-dependent remodeler, ISWI, tends to move nucleosomes from center positions toward the end of DNA[52]. On bulk arrays, the human ISWI ortholog Snf2h evenly spaces nucleosomes along the length of a chromatin fiber[53,54]. If these behaviors are conserved by ISWI on arrays, we would expect remodeling to redistribute the non-nucleosomal DNA across 11 equal linkers in our constructs. Given the flanking free DNA, such actions would drive the 25 bp array toward 30 bp spacing (average 29.7 bp), disfavoring LLPS, and drive the 30 bp array toward 35 bp spacing (average 34.7 bp), favoring LLPS (Fig. 6a, b−cartoon models). To test this model, we mixed a mutant of *Drosophila* ISWI lacking autoinhibitory regulation (2RA) at a 2:1 ratio with the 30 bp arrays in buffers containing 150 mM K$^+$ and 1.5 mM Mg$^{2+}$, where the arrays do not phase separate. ISWI-2RA induces observable phase separation of 30 bp arrays in just 30 minutes, with droplets continuing to grow in size for ~6 hours (Fig. 6a). Similar behavior is observed with wild-type ISWI, which has lower activity due to autoinhibition, but on a longer timescale (Supplementary Fig. 11). In the opposite direction, mixing

25 bp chromatin with the ISWI-2RA mutant for ~5 minutes in the presence of ATP prevents these arrays from phase separating when transferred to physiologic salt. Neither of these changes occurs without ATP or in the presence of the non-hydrolyzable ATP analog AMP-PNP (Fig. 6a, b). Similarly, a catalytically inactive ISWI mutant (K159R) also has no effect on either the 25 bp or 30 bp arrays in the presence of ATP (Fig. 6a, b).

To determine the extent and the heterogeneity of remodeling, we subjected the reactions to MNase digestion, which showed a systematic shift to longer linker lengths in the presence of ISWI + ATP but not ISWI with AMP-PNP, as designed (Fig. 6c, d, Supplementary Fig. 12, Supplementary Data 3). Furthermore, we observe a broadening of the distribution of fragment sizes (Fig. 6c, Supplementary Fig. 12, Supplementary Data 3), consistent with an increase in the heterogeneity of linker lengths, and an increase in fragment sizes smaller than the maximal expected (Fig. 6d, Supplementary Data 3), consistent with incomplete remodeling. Together, these data suggest that incomplete and heterogeneous remodeling is sufficient to induce or disrupt phase separation of the 30 bp and 25 bp arrays, respectively, reflecting the sensitive nature of chromatin condensation to base-pair changes to individual arrays.

We simulated the impact of nucleosome remodeling of chromatin arrays using a Monte Carlo algorithm that captures the stochastic nature of the remodeling process. The algorithm begins with arrays of regularly spaced nucleosomes with either 30 bp or 25 bp linkers, and 15 and 37 bp of bare DNA flanking the terminal nucleosomes at the two ends, respectively. In each cycle of remodeling the nucleosomes in every array are randomly moved in a manner that progressively increases the average linker length toward 35 bp or 30 bp, respectively. These simulations show that just two remodeling cycles are sufficient to disrupt intramolecular nucleosome stacking and enhance inter-fiber interactions in the 30 bp starting arrays, consistent with the experimentally observed induction of phase separation (Fig. 6e). Moreover, as remodeling progresses, the drive to phase separate increases (Fig. 6e). For the 25 bp starting arrays, simulations show again that after just two remodeling cycles the coexistence region shrinks significantly (Fig. 6f). Additionally, the simulations reveal that the heterogeneity introduced by remodeling is insufficient to hinder the decrease in stability of the 25 bp condensates; this occurs because remodeling progressively introduces linker DNA lengths close to 10 N, favoring intra-fiber zig-zag contacts that weaken intermolecular interactions and diminish LLPS.

Together, these data and simulations show that by moving nucleosomes away from or toward 10 N bp positions, such that stacking is favored intermolecularly or intramolecularly, respectively, a remodeler can induce or inhibit phase separation in dynamic fashion. Generalizing, the data suggest that remodelers can control the energetics of higher-order chromatin assembly by changing nucleosome positions.

## Discussion

Interphase metazoan genomes are organized into regions of high chromatin density, on the scale of hundreds of nanometers, connected by regions of low chromatin density, as revealed by a convergence of super-resolution imaging, electron microscopy, and genomic methods[4–6,55–57]. The biophysical mechanisms underlying the emergence, maintenance, and regulation of these compartments are not fully understood[58,59]. It has been proposed that higher-order genome organization is driven by preferential association of chromatin fibers within these domains[60–63], which is determined from a combination of local parameters, such as histone modifications[5,16,64–66] and binding of proteins[66–69] and RNA[70]. Our data suggest that internucleosomal linker length might be an additional local feature that impacts chromatin association and dynamics. We have shown that even small adjustments of one or two base pairs in linker length can have significant effects on

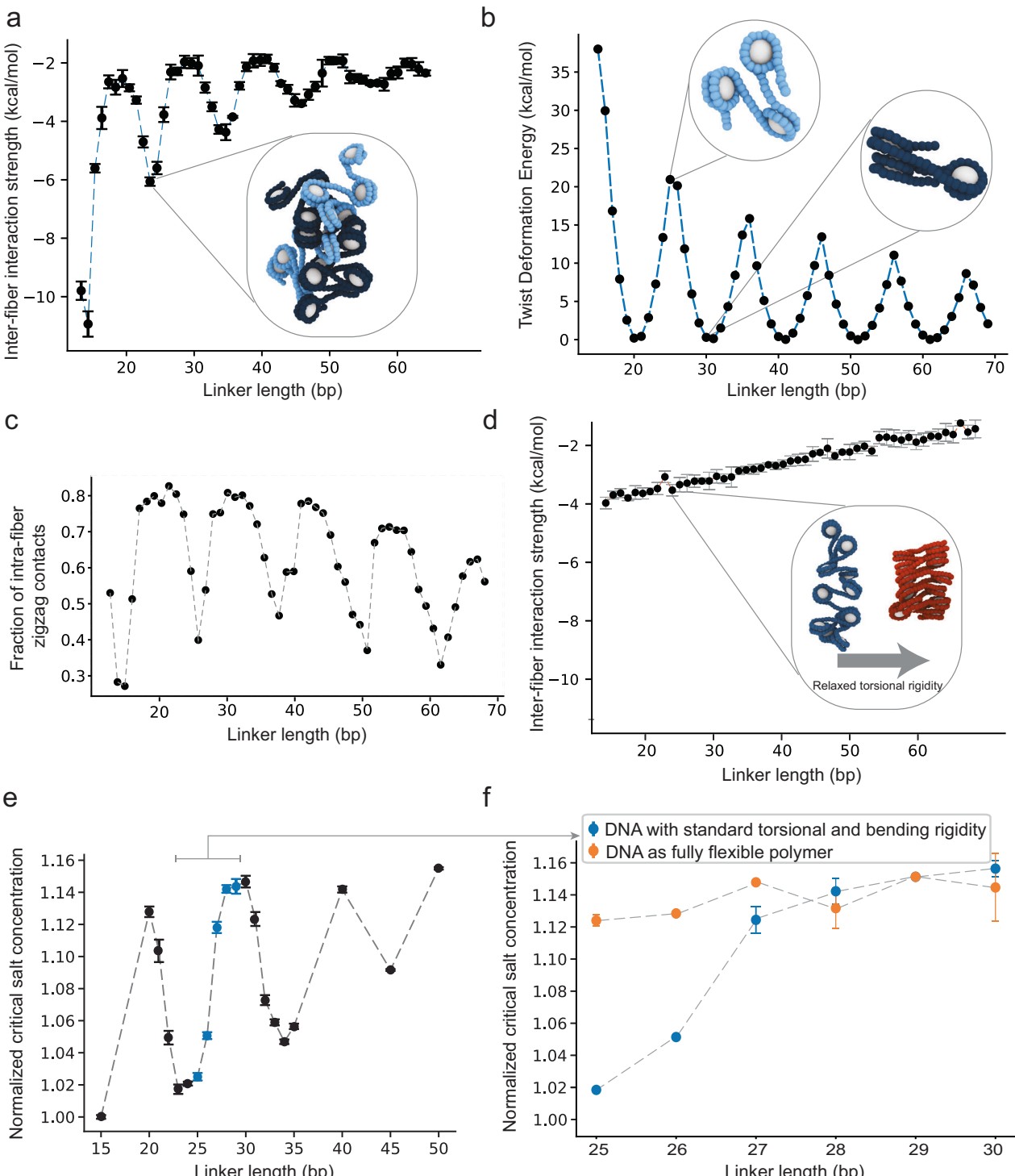

**Fig. 5 | DNA torsional rigidity regulates differences in structure and phase behavior of chromatin with different linkers. a** Binding free energy between two chromatin arrays with indicated linkers, as estimated from the minimum of their PMF wells, as shown in Fig. 4c. $n = 5$ independent repeats. Data are represented as mean ± s.d. **b** Energy cost of twisting the linker DNA to enforce a face-to-face orientation of adjacent nucleosomes. For 10 N + 5 linker lengths, the twist deformation energy is much higher than the gain of making a face-to-face contact (~−4.8 kcal/mol), while for 10 N linkers, the twist deformation energy is negligible. **c** Fraction of intra-fiber zig-zag contacts (i.e., among second nearest neighbors) for chromatin arrays of varying linker lengths. Contacts are normalized by the total number of nucleosomes in the fiber. **d** Inter-fiber binding free energies

as in **a** but for the DNA bending and torsional stiffness set to zero. Inset, illustration of how the structure of a 25 bp chromatin array (left blue) would change if the torsional rigidity of the DNA were removed (right red). $n = 5$ independent repeats. Data are represented as mean ± s.d. **e** Normalized critical salt concentrations taken from the simulated phase diagrams of chromatin solutions with varying linker DNA lengths. Blue points indicate linkers of 25-30 bp. $n = 10,000$ recordings from 100 million timesteps. Data are represented as mean ± s.d. Critical points are calculated by fitting the data to Eqs. (7) and (8), and the error bars are ± error from the least-squares fitting procedure. **f** Normalized critical salt concentrations as in **e** except that DNA bending and torsional stiffness are set to zero (orange points) and compared with the standard case (blue points).

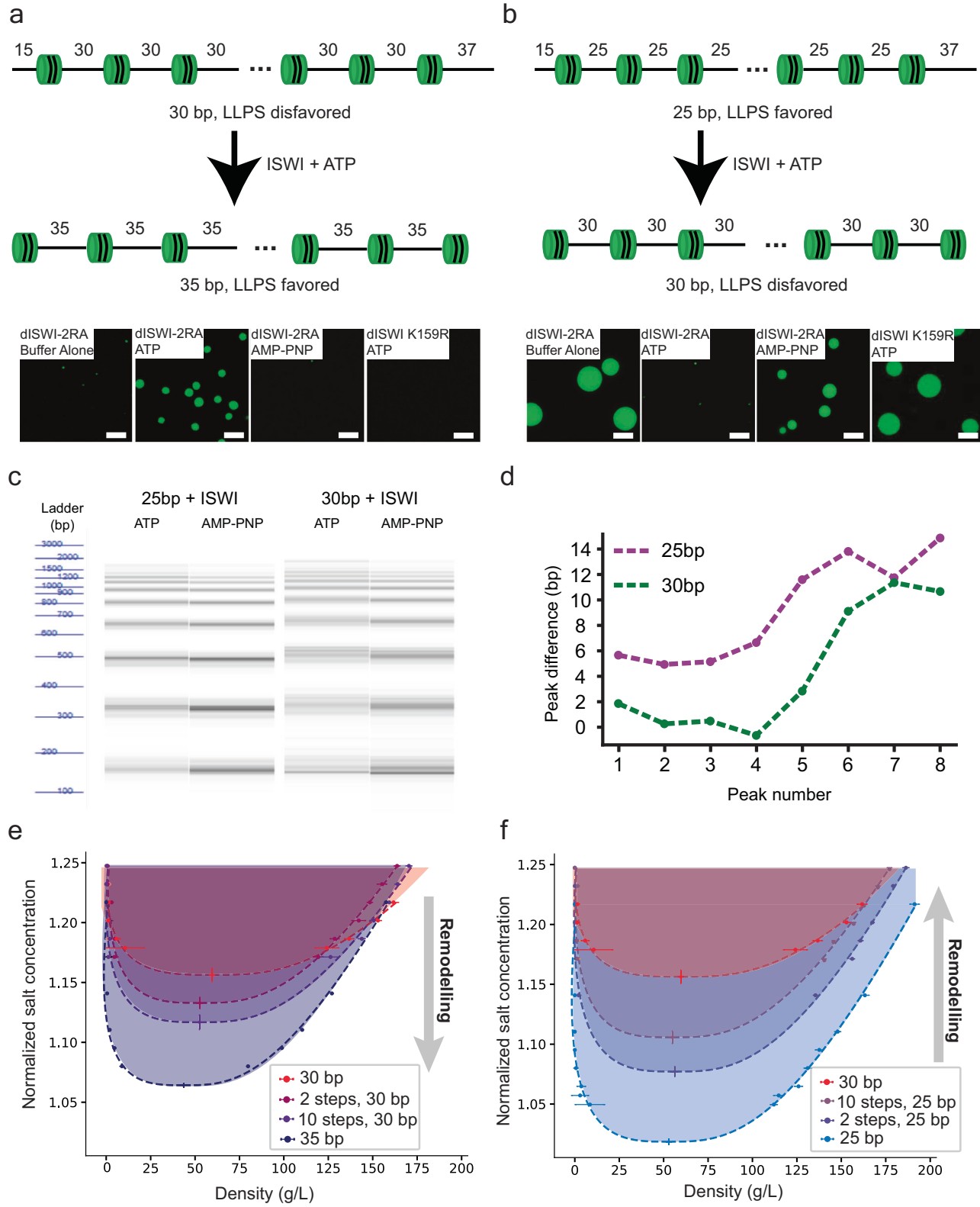

the local structure of the chromatin polymer and consequently the available structural modes of higher-order packing. Furthermore, within the compact regime these small changes can also result in substantially different nucleosome dynamics. Overall, our data suggest that nucleosome spacing, at single base-pair resolution, is an important parameter in considering the structure, dynamics, and interactions of chromatin.

Our study also elucidates an apparent conundrum in chromatin fiber assembly. That is, individual 10 N fibers are more compact and more stable, but they are less able to form condensed mesoscale structures. This is rationalized as a tradeoff between intra-fiber and inter-fiber interactions, where nucleosome stacking within 10 N fibers reduces high-affinity interactions between chromatin chains. By contrast, the more physiologically prevalent 10 N + 5 spacing affords fewer

**Fig. 6 | ISWI regulates chromatin LLPS. a Top** cartoon model of remodeling 30 bp nucleosome arrays to 35 bp arrays by ISWI. **Bottom** representative images of remodeling reactions with conditions indicated after a 6-hour incubation. $n = 3$ independent replicates for all conditions except K159R, where $n = 2$ independent replicates. Scale bar: 10 μm. **b Top** cartoon model of remodeling 25 bp nucleosome arrays to 30 bp arrays by ISWI. **Bottom** representative images of remodeling reactions with conditions indicated after a 6-hour incubation. $n = 3$ independent replicates for all conditions except K159R, where $n = 2$ independent replicates. Scale bar: 10 μm. **c** Bioanalyzer gel-like image of MNase-digested chromatin arrays with indicated conditions, generated from traces in Supplementary Fig. 12. **d** Base-pair difference between the fitted peak positions for the ISWI + ATP condition and the ISWI + AMP-PNP conditions for 25 bp (magenta) and 30 bp (green) starting arrays. **e** Simulation phase diagrams, as in Fig. 1e, for solutions of chromatin arrays with linker lengths of 30 bp (pre-remodeling), at two intermediate stages of the remodeling process (2 cycles and 10 cycles), and after full remodeling when all fibers are transformed into 35 bp arrays (post-remodeling). $n = 10,000$ recordings from 100 million timesteps. Critical points are calculated by fitting the data to Eqs. (7) and (8), and the error bars are ± error from the least-squares fitting procedure. **f** Remodeling progression as in e, but for a solution of 25 bp arrays remodeled to 30 bp spacing.

intra-fiber contacts, enabling strong interactions between chains and higher-order condensation. This comparison highlights the effective opposition of local folding and global compaction.

Because each base pair of DNA accounts for a 35° rotation, small changes in linker length cause large changes in the relative orientation of successive nucleosomes. Therefore, factors that bind to multiple adjacent nucleosomes simultaneously[71–74] likely exhibit different activities toward different chromatin substrates. For example, a cryo-EM study demonstrated that Polycomb repressive complex 2 (PRC2) can assemble on dinucleosomes, which are important substrates for H3K27me3 methylation and spreading[71,75]. The binding contacts were observed to be different between 10 N (30 bp and 40 bp) and 10 N + 5 (35 bp) dinucleosomes, and the DNA was more bent in the latter case. These changes likely affect binding affinity and functional output, which might explain the greater enzymatic activity of PRC2 on 20 bp dinucleosomes than on 46 bp and 66 bp dinucleosomes[76]. Similarly, CENP-N promotes stacking of 20 bp chromatin into 30 nm fibers[77]; the affinity of this interaction is likely decreased for arrays with 10 N + 5 spacing, where such compaction would induce substantial torsional strain. Compensatory changes in linker length (e.g., 25 + 2 bp and 25 − 2 bp) of tri-nucleosomal units may also enable the formation of local stacks that facilitate binding regardless of the context of the surrounding array. Thus, in general, oligonucleosome configurations can create short stacking motifs or open nucleosome faces, influencing regional dynamics and activity of factors with preferential interactions. Superimposed upon these local considerations is that in higher-order chromatin assemblies, linker length will also determine the frequency and strength of inter-fiber interactions. These interactions may compete with or complement binding to regulatory factors. Future studies of biochemical activity within chromatin droplets produced with different linkers will shed light on these issues and their potential importance to chromatin function in the cell.

Since interactions between chromatin fibers are dependent on linker length, nucleosome remodelers can alter not only linear nucleosome positions but also higher-order chromatin assembly and dynamics. We demonstrated such behavior here in the context of phase separation of synthetic arrays, but the concept should hold for cellular chromatin as well. Moreover, most remodelers are equipped with accessory domains that recognize specific nucleosomal contexts, such as epigenetic modifications, histone variants, and locally concentrated transcription factors[78]. Our data support a model wherein remodelers do not merely impart energy or DNA accessibility to specific linear regions of chromatin characterized by such features, but also affect global chromatin structure, potentially in switch-like fashion. Further studies of remodeler actions on higher-order chromatin structure and dynamics will be needed to understand the importance of such effects on genome function.

Nucleosome positioning in cells is determined by myriad factors, including remodelers, DNA sequence, transcription activity, and protein binding[79]. Thus, patterns of linker lengths vary across the genome. *Drosophila* chromatin, when divided into 9 chromatin states (e.g., active promoter, heterochromatin, etc.), exhibits varying degrees of array regularity, i.e., constant linkers across a region[80,81]. Of the chromatin states with regular arrays, the average linker length differs by only a few base pairs. But our data suggest that these differences could have significant effects on chromatin structure and dynamics. For example, Polycomb repressed domains have an average linker length of 26 bp, which should stably interact with neighboring fibers and have slow dynamics, whereas transcription elongation regions have an average linker length of 20 bp, which should interact with neighbors more weakly and move more readily. Another study clustered linker lengths across the genome into seven groups[82]. Of the four groups that exhibited regular nucleosomes, two fall into the 10 N class (39 bp and 40 bp), and two fall into the 10 N + 5 class (25 bp and 46 bp). There is marked heterogeneity within each group. However, the groups persist across epigenetic states, but with different proportions, suggesting they are functionally relevant. Notably, satellite DNA is enriched for the 10 N + 5 class and strongly depleted of the 10 N class. These studies suggest that linker length patterns may be selected for certain parts of the genome, perhaps reflecting functionally important differences in compaction and dynamics. Consistent with this idea, although most nucleosomes in metazoans are poorly positioned[79,80,83], nucleosome spacing is more regular, suggesting more stringent constraints on linker length compared to nucleosome positioning per se[84]. Thus, regulation of nucleosome spacing, and consequently higher-order assembly and dynamics of chromatin, may be an important means of controlling functions of the genome.

## Methods

### Cloning of bacterial expression vectors containing 12×601 with linker lengths 25 bp – 30 bp

Six expression vectors containing 12 repeats of the Widom 601 sequence[28] separated by intervening sequence lengths of 25 bp – 30 bp were cloned using the same method. The 12×601 array sequence was split into 4 fragments each, which were purchased from Genscript. The fragments were joined by ligating unique complementary overhang sequences generated by BsaI digestion. Joined 12×601 sequences were inserted into the pWM vector[38] through HindIII and BamHI sites.

The fragments purchased from Genscript generated arrays with 15 bp DNA flanking the 12×601 sequence (30 bp total). For remodeling assays, we extended one end to 37 bp (52 bp total) by ligating PCR-amplified pWM vector with gel-purified 12×601 digested by EcoRV-HF (NEB). The PCR primers were CTCGAGGAAGACATCCCCTGATATCCAACTCAGATCAAGCTTGGG-CGTAATCATAGT, and ATCGGATCCCCGGGTAC.

The sequences of all eight arrays can be found in Supplementary Data 1.

### Preparation of 12×601 DNA arrays for chromatin assembly

Large-scale purification and digestion of all 12×601 arrays were performed as previously described[38]. Briefly, pWM plasmids containing 12×601 were purified from a 6 L culture of transformed *dam⁻/dcm⁻ E. coli* (NEB) using Plasmid Giga Kit (Qiagen). The 12×601 arrays were separated from the vector by EcoRV-HF (NEB) digestion. 12×601 arrays and the digested vector were purified by phenol-chloroform

extraction and precipitation without further separation as the vector served as "carrier" DNA in the chromatin assembly procedure below.

## Preparation of histone dimers and octamers

Expression and purification of *H. sapiens* histone H2A, H2B T116C, H3 C111A, and H4 were performed as previously described[38]. Histone H2B T116C was labeled with Alexa Fluor 594 (ThermoFisher) as previously described[38]. Histone octamers were assembled with mixtures of histone H2A, H3 C111A, H4, and H2B T116C, with and without Alexa Fluor 594, and were purified by size exclusion chromatography as previously described[38]. Histone H2A/H2B dimers, due to excess molar ratio of histone H2A and histone H2B in the octamer assembly, were also saved from chromatography fractions. Purified histone octamers and dimers were aliquoted, flash frozen, and stored at −80 °C. The concentrations of each batch of histone octamers were empirically determined by assembling mononucleosomes on a single Widom 601 sequence using dialysis under continuously decreasing salt concentration[39]. The concentrations of histone H2A/H2B dimers were calculated from absorbance at 280 nm and their molar extinction coefficient ($11920 \, M^{-1} \, cm^{-1}$).

## Assembly of 12×601 nucleosome arrays

12×601 DNA arrays with "carrier" vector DNA were mixed with 1% fluorophore-labeled histone octamers and histone H2A/H2B dimers at a molar ratio of 1:1.3:0.2 for 601 positioning sequence:octamer:dimer. The excess of octamer relative to positioning sequence and the addition of dimer aid in the complete assembly of nucleosomes, and over-assembly is prevented by the presence of "carrier" DNA. Nucleosome formation onto the 601 sequences during dialysis under continuously decreasing salt concentration, purification by sucrose gradients, and concentration quantification were performed as previously described[38,39]. Assembly quality was determined by running an electrophoretic mobility shift assay on mononucleosomes formed from digesting dodecameric arrays by EcoRI-HF (NEB)[38,39].

## ISWI purification

Plasmids for all dISWI constructs were synthesized by Twist Biosciences. ISWI genes were fused to a C-terminal tandem intein and chitin-binding domain[85] and inserted into the pET-29b(+) plasmid between the NdeI and XhoI sites. All dISWI constructs were purified by the same procedure. In each case, *E. coli* strain Rosetta(DE3) was transformed with a dISWI plasmid and grown in LB media at 37 °C until an OD600 of 0.3. Media was then transferred to 16 °C for 1 hour before inducing protein expression with 1 mM IPTG and allowed to grow overnight. Cells were then pelleted by centrifugation for 30 minutes at 6000 x *g* and resuspended in lysis buffer (40 mM Tris-HCl, pH 7.5, 1 M NaCl, 1 mM EDTA, 5% glycerol, 0.1% Triton X-100, and cOmplete Protease Inhibitor Cocktail (Roche)). All following steps were performed at 4 °C. Cells were then lysed by a freeze/thaw cycle followed by sonication. The lysed cells were then centrifuged at 30,000 x *g* for 30 minutes and the supernatant was recovered. The clarified lysate was then incubated with Chitin Resin (NEB) with rocking for 2 hours. The slurry was then placed in a gravity column and the flow-through was discarded. The resin was then washed with 1 L of wash buffer (40 mM Tris-HCl, pH 7.5, 1 M NaCl, 1 mM EDTA, 5% glycerol). As the very last of the wash buffer passed through the column, 40 mL of elution buffer (40 mM Tris-HCl, pH 7.5, 1 M NaCl, 1 mM EDTA, 5% glycerol, 50 mM DTT) was added without disturbing the resin and 10 mL of elution buffer was allowed to flow through before stopping the flow and incubating overnight. The next day, 1 mL fractions were eluted from the column and analyzed by SDS-PAGE. Protein-containing fractions were then pooled and dialyzed into storage buffer (40 mM Tris-HCl, pH 7.5, 150 mM KOAc, 1 mM EDTA, 50% glycerol, 1 mM DTT). Following dialysis, fractions were flash frozen in liquid nitrogen and stored at −80 °C.

## Turbidity assay

Absorbance at 300 nm was used to monitor the scattering of light by phase-separated droplets with higher refractive indices than the surrounding buffer and sizes ~300 nm or larger. By monitoring the presence of droplets as a function of salt concentration, we identified the threshold concentration for phase separation, which can be compared between chromatin constructs.

Nucleosome arrays were first diluted to 83 nM in Chromatin Dilution Buffer (25 mM Tris-OAc, pH 7.5, 0.1 mM EDTA, 5 mM DTT, 0.1 mg/mL BSA, 5% glycerol). Diluted nucleosome arrays were mixed with 2x Phase Separation Buffer (25 mM Tris-OAc, pH 7.5, 0.1 mM EDTA, 5 mM DTT, 0.1 mg/mL BSA, 5% glycerol, 2 mg/mL glucose oxidase, 350 ng/mL catalase, 4 mM glucose, 2X indicated mM of KOAc, pH 7.5, 2X indicated mM of Mg(OAc)$_2$, pH 7.5) by pipetting 10 times. After a 30-second incubation, 3 μL of the mixture was applied to the pedestal of a NanoDrop One$^C$ instrument (Thermo Scientific), and absorbance was measured at 300 nm.

Threshold concentration was calculated by linear interpolation of two points flanking 0.2 absorbance using the equation: threshold concentration (mM) $= \frac{x_0(y_1 - 0.2) + x_1(0.2 - y_0)}{y_1 - y_0}$, where $(x_0, y_0)$ and $(x_1, y_1)$ are pairs of points whose absorbance is below and above 0.2, respectively. Multiple repeats were individually fitted, and the interpolated result with standard deviation is reported. Alternatively, we fitted a logistic function to all the points and used the fitted curve to interpolate the concentration of salt at which absorbance equals 0.2. Both methods result in threshold concentrations within 1–2 mM K$^+$ (<1%) of each other, and we favor the linear method here.

## Microscopy

FRAP (Fig. 2, Supplementary Figs. 4 and 5) and confocal fluorescence images (Supplementary Fig. 1) were recorded on a Leica SP8 confocal microscope with a 552 nm wavelength laser, a 20×0.4 NA dry objective (FRAP) or a 63 × 1.4 NA oil immersion objective (confocal), and a hybrid (HyD) detector.

ISWI reactions (Fig. 6) were imaged on a Nikon AX confocal microscope using a 60 × 1.4 NA oil immersion objective and a 488 nm solid state laser.

## Phase separation for microscopy

To assess phase separation, nucleosome arrays were diluted with Chromatin Dilution Buffer and then mixed with 2X Phase Separation Buffer at the indicated salt concentration to a final volume of 20 μL. The final nucleosome array concentration was 83 nM for FRAP assays (Fig. 2, Supplementary Figs. 4 and 5), 68.5 nM for ISWI assays (Fig. 6), and 43 nM otherwise (Supplementary Fig. 1). Phase-separated chromatin was then transferred to mPEGylated 384-well microscopy plates, prepared as previously described[38].

## Fluorescence recovery after photobleaching (FRAP)

After at least 2 hours of incubation, a circular region of interest in the center of the condensate, less than 1/3 in diameter relative to the whole droplet, was bleached by high-intensity illumination of a 552 nm wavelength laser, such that the fluorescence intensity fell to 20-30% of the original. Images were taken at defined intervals and were analyzed using Fiji. Bleached droplets were selected individually based on their size, bleached region (i.e., bleaching occurred in the center of the droplet and did not occur on the droplet surface), and a lack of detectable fusion events during acquisition, and were aligned across time. Fluorescence recovery was calculated as the ratio of fluorescence intensity of the bleached region relative to the whole droplet, which quantifies the rate and degree of internal mixing while accounting for photobleaching. The ratio was normalized such that the fluorescence recovery for the first post-bleach image is 0 by the equation:

normalized fluorescence recovery$(t) = \frac{I_t - I_o}{1 - I_o}$, where $I_t$ is the intensity ratio at time $t$ and $I_o$ is the intensity ratio of the first post-bleach image.

The normalized fluorescence recovery was fitted to a one-phase association curve using Prism (GraphPad) with the equation:

normalized fluorescence recovery$(t)$
= normalized fluorescence recovery$(0)$
+ (plateau − normalized fluorescence recovery$(0)$) × $\left(1 - e^{-kt}\right)$.

### Nucleosome remodeling assays

**For imaging.** dISWI reactions were initially prepared in 10 µL volume in buffer containing 25 mM Tris-OAc, pH 7.5, 1.5 mM Mg(OAc)$_2$, 0.1 mM EDTA, 5 mM DTT, 0.1 mg/ml BSA, 5% glycerol, and 1 mM ATP (or 1 mM AMP-PNP if specified) with 137 nM dISWI, dISWI-2RA, or dISWI-K159R. After approximately 5 minutes, additional buffer was added, raising the final volume to 20 µL and adding 150 mM KOAc while holding all other buffer conditions the same. Reactions were then transferred to a 384-well imaging plate and representative images were acquired after 6 hours.

**For linker length determination.** 50 nM nucleosome arrays and 800 nM ISWI-2RA were diluted in 20 µL buffer containing 25 mM Tris-OAc, pH 7.5, 3 mM Mg(OAc)$_2$, 5 mM DTT, 0.1 mg/ml BSA, 5% glycerol, and either 2 mM ATP or 2 mM AMP-PNP, and incubated for 2 hours at room temperature. 1 µL of MNase (NEB) was added to the 25 bp array (0.2 gel unit) and the 30 bp array (0.15 gel unit). MNase activity was initiated with 0.6 µL of CaCl$_2$ to arrive at a final concentration of 3 mM. Reactions were terminated after a 15-minute incubation at 37 °C by the addition of 60 mM EDTA and were extracted by phenol/chloroform/isoamyl alcohol. The aqueous phase was run on agarose gels and Bioanalyzer for fragment visualization and analysis.

### Modeling

Our chromatin coarse-grained model integrates information at three complementary resolutions: (a) atomistic data of nucleosomes, histone proteins, and DNA, (b) a DNA and amino acid sequence-dependent model that represents histone proteins at residue resolution and DNA at the base-pair level, which we term the "chemical-specific coarse-grained chromatin model", and (c) a minimal chromatin model that represents nucleosomes with just ~30 particles yet it realistically captures the mechanical rigidity of the DNA, its modulation with length, and the balance between such rigidity and the anisotropic strength of nucleosome-nucleosome interactions. Nucleosome-nucleosome and nucleosome-DNA interactions in the minimal model are modeled with orientationally-dependent potentials fitted to reproduce internucleosome potentials of mean force calculated with our chemically-specific coarse-grained chromatin model. The crucial mechanical rigidity of the DNA is described with a coarser version of the rigid-base-pair model parameterized based on extensive simulations of DNA at the chemical-specific resolution, as described in ref. 50.

For this work, we have developed an updated version of our minimal coarse-grained chromatin model first presented in ref. 50, which allows us to consider any DNA linker length. Our minimal coarse-grained chromatin model has a resolution of one bead per five DNA base pairs and one bead for the nucleosome histone core. The DNA is categorized into either nucleosomal DNA, which is rigidly fixed to the nucleosome core or linker DNA which is free to move and connected via bonds. Each bead is a spherical rigid body described by a position vector $\vec{v}$ and a unit quaternion $\vec{q}$. The DNA bonded interaction uses a Rigid Base-Pair like potential[86–88], which has a

functional form of

$$E_{minimal-RBP} = \frac{1}{2}\Delta\vec{\phi}K\vec{\phi} \tag{1}$$

$$\Delta\vec{\phi} = \left(\vec{\phi} - \vec{\phi_0}\right) \tag{2}$$

$$\vec{\phi} = (shift, slide, rise, tilt, roll, twist) \tag{3}$$

Where $\vec{\phi}$ is a 6-dimensional vector of the instantaneous helical parameters between two DNA beads composed of three angles: tilt, roll, twist; and three distances: shift, slide, rise. $\vec{\phi_0}$ is the equilibrium value of the helical parameters and $K$ is the 6×6 stiffness matrix. The values of $\phi_0$ and $K$ were parameterized from MD simulations (reported in ref. 50) of 200 bp strands of DNA using the standard base-pair resolution RBP model with parameters from the NAFlex webserver[89]. The values we use in this work are as follows: The equilibrium rise between base pairs is 16.45A, the equilibrium twist is 172° and the equilibrium values of the other helical parameters are zero. We set $K$ to be a diagonal stiffness matrix with diagonal values

$$K_{diag} = (0.301, 0.235, 1.56, 0.00614, 0.00515, 0.00724) \tag{4}$$

The pairwise interactions are shifted and truncated Lennard-Jones potentials. The DNA-DNA interactions are repulsive, approximating the electrostatic repulsion of negatively charged DNA. The DNA-Core interactions are attractive, approximating the electrostatic attraction between DNA and the histone core proteins. The pair potentials are empirically fitted to reproduce the orientation dependence of Nucleosome-Nucleosome interactions as explained in[50]. Furthermore, we have an empirical mapping from monovalent salt concentration to the pair potential parameters.

$E_{Minimal-LJ} = E_{LJ}(r_c)$ for $r < r_c$ ; 0 otherwise

$$E_{LJ}(r) = 4\epsilon\left[\left(\frac{\sigma}{r}\right)^{12} - \left(\frac{\sigma}{r}\right)^{6}\right] \tag{5}$$

where $\epsilon$ is the interaction strength, $\sigma$ is the zero crossing point, $r$ is the distance between the pair of interacting particles, and $r_c$ is the cutoff distance. The simulations in this work were performed at different salt concentrations with parameters given in ref. 50.

The total energy of the model is then the sum of the bonded and pairwise terms:

$$E = E_{minimal-LJ} + E_{minimal-RBP} \tag{6}$$

The model is simulated using Langevin dynamics, which implicitly includes the effects of solvent using a mean-field approximation. A modified version of LAMMPS[90] was used for all simulations, as described in ref. 50.

The mean-field approximation assumes that varying the salt concentration only modulates the screening of the mean electrostatic potential stemming from the chromatin charges. Such an approximation is crucial to reduce the dimensionality of the phase space of a chromatin solution and enables us to investigate LLPS. However, the approximation ignores effects that might be important modulators of chromatin phase separation, including ion correlation and polarization effects, ion release upon formation of inter-chromatin interactions, the differential residence times of diverse ions on specific amino acids and nucleotides leading to heterogeneous ionic distributions, and cross-linking of species. While these effects are expected to be modest in solutions containing only monovalent ions at physiologic conditions, they are expected to become significant at higher concentrations or when polyvalent ions are present.

**Creating initial structures.** In our previous work, we only used linker DNA lengths that were a multiple of 5 to fit easily with the coarse-grained representation of 1 bead per 5 DNA base pairs. In this work, we use the model for any linker DNA length. We do this using the following method: First we use the chemically specific model (described in[50]) to create an initial chromatin structure with the desired linker DNA length at 1 bead per base-pair resolution, we then map the structure to the minimal model representation by grouping every sets of 5 DNA base pairs into one bead. When we reach the very end of the chromatin, we neglect any last base pairs that remain.

**Single fiber simulation details.** For all single fiber simulations in this work, we used temperature replica exchange molecular dynamics (T-REMD) with replicas spanning a temperature range of 300–600 K. Depending on the system size, we used an appropriate number of replicas to give exchange probabilities close to 0.3. The simulations were run for at least 200 million timesteps, with T-REMD exchanges attempted every 100 timesteps. Coordinate snapshots were recorded every 100 thousand timesteps. The first half of the trajectories were neglected from the analysis. Only the 300 K replicas were used for analysis.

**PMF calculations.** We computed the PMFs using umbrella sampling with the COLVARS package[91]. Additionally, we used T-REMD for each umbrella window to enhance the sampling. This was necessary due to the large number of different chromatin configurations that can occur for the same value of the collective variable. Importantly, the T-REMD was done independently for each window, i.e., no exchanges occurred between different windows, simply each window was run using a temperature replica exchange scheme with 16 replicas spanning from 300–600 K. The standard Metropolis acceptance criteria can be used as long as we are careful to add the biasing potential to the total potential energy of the system.

For the inter-fiber chromatin PMFs, the collective variable was biased using a harmonic potential with a force constant of $0.002$ kcal/mol/Å$^2$ and 16 equally spaced umbrella windows were used spanning 0–600 Å. For the intra-fiber chromatin interactions, the collective variable was biased used a harmonic potential with a force constant of $0.0005$ kcal/mol/Å$^2$. We used 20 windows spanning 0 Å to 950 Å. Each window was run for 10 million timesteps. The PMFs were computed from the umbrella windows using WHAM[92]. Five independent repeats were done; in the main figures, the plotted PMFs are the mean of these five repeats, and the shaded regions are the standard deviation of the five repeats.

**Direct coexistence simulations.** To compute the phase diagram of 12-nucleosome chromatin, we employ the direct coexistence method[93–95] using 125 independent 12-nucleosome chromatin arrays at different salt concentrations. The chromatin array initial structures are obtained by generating them from T-REMD simulations at the corresponding salt concentration.

In a direct coexistence simulation, we place both phases, i.e., the dilute liquid and the condensed liquid phase, in the same elongated simulation box. The simulation is performed until both phases reach equilibrium at their coexistence densities. Once equilibrium is reached, we measure the densities of coexistence by computing an average density profile along the long side of the simulation box with the center of mass fixed. Once equilibrated, the simulations were run for approximately 100 million timesteps, and coordinate snapshots were recorded every 10,000 timesteps. This is greater than the correlation time of our model, which is equal to 4000 timesteps (1 timestep = 100 fs). The simulation box dimensions were 1200 Å x 1200 Å x 5000 Å.

We estimate the critical salt concentration $c_c$ by fitting the density difference between the low-density phase, $\rho_{l(c)}$, and the high-density

phase, $\rho_{h(c)}$, to the expression given below[96],

$$\left( \rho_{h(c)} - \rho_{l(c)} \right)^{3.06} = \left( 1 - \frac{c}{c_c} \right) \tag{7}$$

where $d$ is a fitting parameter. The critical density $\rho_c$ is then estimated using the law of rectilinear diameter,

$$\frac{1}{2} \left( \rho_{l(c)} + \rho_{h(c)} \right) = \rho_c + s(c_c - c) \tag{8}$$

where $s$ is a fitting parameter.

The coexistence curves separate the region of parameter space where a chromatin solution is found as a single homogeneous phase (no LLPS) from where it demixes into two phases (LLPS). To obtain them, we perform different direct coexistence simulations, each at a different salt concentration. For salt concentrations where we can detect two distinct phases in equilibrium, we measure the coexistence densities of these two phases and plot them to build the coexistence curve. If, in contrast, we observe the formation of a single homogeneous phase, we determine that such a salt concentration is below the critical salt concentration.

**Contact analysis.** Contact analysis on the output trajectories from the direct coexistence simulations was conducted to determine the types of interactions between nucleosomes within chromatin fibers. The analysis, performed on python scripting in Ovito API[97], focused on identifying and quantifying different types of nucleosome-nucleosome contacts, categorized as face-to-face, side-to-side, and face-to-side interactions.

The data pipeline was configured to create bonds between particles using a modifier with a cutoff distance of 120 units, to define the bond topology necessary for subsequent contact analysis. Four quaternion components were extracted to determine the orientation of each nucleosome. Contacts between nucleosomes were then classified into three types based on their relative orientations and positions: face-to-face, where the angle between the Z-axis vectors of two nucleosomes was less than 45 degrees, side-to-side, where the distance between the centers of two nucleosomes was less than the cutoff distance, and their Z-axis vectors were perpendicular, and finally face-to-side, where one nucleosome's Z-axis vector was perpendicular to the vector connecting the centers of two nucleosomes.

The mean and standard deviation for each contact type and total bonds were computed across all frames and fibers. These statistics provided insights into the average number of each contact type and the variability in the contact numbers. This method allowed for a detailed analysis of nucleosome interactions and their structural organization within the chromatin fiber, providing insights into the mechanisms underlying chromatin organization and dynamics.

**Remodeler simulations.** We simulated the impact of nucleosome remodeling of chromatin arrays within condensates by preparing chromatin solutions at different stages of the remodeling process. For this, we designed a Monte Carlo algorithm that captures the stochastic nature of the remodeling process—e.g. stochastic binding of ISWI to one of the many arrays in the condensate. The algorithm chooses one fiber at random and enforces the effect of one ISWI molecule remodeling all its nucleosomes sequentially before unbinding. Remodeling is defined in our algorithm as equalizing the DNA length on both the entry and exit sides of a given nucleosome. Consequently, the algorithm starts with a solution where all chromatin arrays have the same linker DNA lengths (e.g., either all 30 bp or all 25 bp with 15 bp and 37 bp, respectively, of DNA flanking the terminal nucleosomes on the two ends, to parallel the experimental system) and progressively produces heterogeneous mixtures of chromatin arrays with diverse

distributions of linker DNA lengths. Eventually, after many iterations, the linker DNA lengths even out again, and the algorithm reaches the state of full remodeling, where all fibers have the same homogeneous distribution of DNA linker lengths (e.g., ~35 bp for the systems starting as 30-bp arrays, or ~30 bp for the systems starting as 25 bp arrays). More specifically, the algorithm defines one remodeling cycle using the following steps: (1) One chromatin fiber is chosen at random; (2) If the fiber contains flanking DNA beyond the terminal nucleosomes, the first and/or last nucleosomes are pushed to the ends of the fiber until no flanking DNAs are left; (3) One nucleosome between 2 and 11 is chosen at random; (4) The lengths of the entering and exiting linker DNAs of the chosen nucleosome are equalized; (5) The adjacent nucleosomes are remodeled next (e.g., if the 5th nucleosome was chosen in step 4, the 4th and the 6th nucleosomes are then remodeled in step 5), and step 4 is repeated until both ends of the fiber are reached.

**Structure variability analysis.** To analyze the structural similarity of chromatin arrays, we employed K-means clustering based on the distance root mean square deviation (dRMSD). This method allowed us to group similar structures and identify common conformational states within the chromatin simulations. The goal of K-means clustering is to minimize the variance of a distance metric within each cluster, starting with a fixed number of clusters.

The clustering analysis was conducted on AmberTools[98,99], on the previously obtained trajectory data, based on the positions and orientations of the histone cores. The clustering algorithm was initialized with 20 clusters, using a random point selection method. The algorithm was set to iterate up to a maximum of 1000 iterations to ensure convergence.

The dRMSD values between structures were used as a distance metric, computed as part of the clustering process to quantify structural similarity. These values served as the basis for the K-means method, ensuring that structures within the same cluster had minimal conformational deviations.

To compute the dRMSD between structures, let $\vec{x}_i$ and $\vec{y}_j$ be the positions of the i-th nucleosome of fiber 1, and the j-th nucleosome of fiber 2. The dRMSD between the two fibers is then computed as

$$dRMSD = \sqrt{\frac{2}{N(N-1)} \sum_{i<j} ||\vec{x}_i - \vec{y}_j||} \qquad (9)$$

The main advantage of using dRMSD over RMSD in this instance is that it is a metric independent of a reference structure, and allows for an unbiased clustering method.

## Statistics and reproducibility
Biochemical experiments were performed under consistent conditions, and the results were replicated across many experimenters and institutions. Descriptions of the data, including sample sizes and error representations, are associated with each figure. No statistical method was used to predetermine sample size. For FRAP experiments, droplets with bleach spots near the surface and droplets that fused with other droplets were excluded from the analysis. The first 20 million timesteps of simulations were discarded to ensure proper thermalization and convergence of computational simulations. No other data were excluded from the analyses. Initial configurations for simulations were generated using randomized starting positions and velocities to ensure unbiased sampling. The biochemical experiments were not randomized. The investigators were not blinded to allocation during experiments and outcome assessment.

## Reporting summary
Further information on research design is available in the Nature Portfolio Reporting Summary linked to this article.

## Data availability
Source data for in vitro results are provided as a Source Data file. The computational datasets generated during the current study are stored on https://doi.org/10.6084/m9.figshare.28672877. Source data are provided with this paper.

## Code availability
The authors are delighted to share the computational implementation of their models with the community, which can be found at https://doi.org/10.5281/zenodo.15351283. The authors are happy to answer any questions and comments by email.

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

## Acknowledgements

We thank Daniel and David Isenberg for their generous support of the MBL Chromatin Collaborative in honor of their father, Irvin Isenberg, and the other members of the Collaborative for critical insights and discussion. We thank Jerelle A. Joseph and Esmae J. Woods for interesting discussions on the simulation aspects of this work. Research was supported by the Howard Hughes Medical Institute, a Paul G. Allen Frontiers Distinguished Investigator Award (to M.K.R.), grants from the Welch Foundation (I-1544 to M.K.R.), and the National Institutes of Health (R35GM141736 to M.K.R.; 5R35GM147477-03 to S.R.). This project made use of time on high-performance computing granted via the UK High-End Computing Consortium for Biomolecular Simulation, HECBioSim (http://hecbiosim.ac.uk), supported by the Engineering & Physical Sciences Research Council (EPSRC) (grant no. EP/R029407/1 to R.C.G.). J. H. acknowledges funding from the Herchel Smith Postdoctoral Fellowship Fund and the UK Research and Innovation (UKRI) EPSRC under the UK Government's guarantee scheme (EP/X02332X/1 to J.H.), following funding by the European Union's Horizon 2020 Marie Skłodowska-Curie Actions (MSCA) Fellowship programme. R.C.-G. and M.J.M. acknowledge funding from the European Research Council (ERC) under the European Union Horizon 2020 research and innovation programme (Starting Grant 803326 'InsideChromatin' to R.C.-G.) and the UKRI EPSRC under the UK Government's guarantee scheme (EP/Z002028/1 to R.C.-G.), following funding by the ERC (Consolidator Grant 'ChromatinDroplets') under the European Union's Horizon Europe research and innovation programme.

## Author contributions

Conceived the study, L.C., R.C.-G., B.A.G., S.R., M.K.R. Performed biochemical experiments, L.C., J.L., L.K.D. Performed computer simulations, M.J.M., S.E.F., J.R.E., J.H. Drafted manuscript, L.C., M.K.R. Revised manuscript, all authors

## Competing interests

The authors declare no competing interests.
