## [Peer Review file · Nature Communications]

Nucleosome Spacing Can Fine-Tune Higher Order Chromatin Assembly

Corresponding Author: Professor Michael Rosen

Version 0:

Reviewer comments:

Reviewer #1

(Remarks to the Author)

In this study experimentalists (Rosen lab) and computational scientists (Colleparado-Guevara lab) join forces to study liquid condensates formed from chromatin arrays. The combination of in vitro experiments and computer simulations of these two groups, who have worked on this system for several years, resulted in a manuscript that provides insight into liquid-liquid phase separation in unprecedented detail. I consider this work as a major advance in the field of chromatin research, particularly in terms of connecting small scales (nucleosomes) to larger scales (liquid condensates inside the nucleus). This is crucial because the intermediate structure, the classical chromatin fiber, still found in standard textbooks, has been understood as an artefact of in vitro experiments. We still do not have a good idea how to connect these two scales, and the current work shows how details at the small scales, in particular the DNA-linker length between nucleosomes, affect the propensity of nucleosome arrays to either stay more local or merge into a "nucleosome melt". To my knowledge, there is currently no study that shows the mechanism behind this phenomenon so clearly. I strongly recommend publishing this insightful study in Nature Communications.

The manuscript is very clearly written and I have only a few minor comments:

- (1) First paragraph in the introduction: "local diffusion" of nucleosomes, please specify whether you mean in space or along the DNA molecule (I assume you mean in space).
- (2) On page 7, the authors first refer to a Fig. 1d that does not seem to quite match the actual figure and then to Fig. 1e, which seems not to exist. Please check the numbering.
- (3) I did not fully understand the description of the computational algorithm to mimic remodeling (page 42), especially point 5. What is meant by the "sequential nucleosome neighbour", especially in the case when there are two free flanking sequences at the start.

(Remarks on code availability)

Reviewer #2

(Remarks to the Author)

This manuscript reports how chromatin phase separation depends on the linker length between nucleosomes. The model systems are based on 12-mer nucleosome arrays with controlled linker lengths ranging from 25-30 bp. The main conclusion is that the relative rotational positioning of one nucleosome particle to the next plays a critical role in regulating chromatin phase separation. They carried out MD simulations to suggest that this effect is due to the relative extent of nucleosome stacking in an intra- vs inter-array fashion. According to the results, 10N+5 linkers will make intra-array nucleosome stacking less efficient than 10N linkers, leading to more efficient inter-array intercalation, and consequently, facilitating phase separation. They also added ISWI to show how chromatin remodeling impacts phase separation and may regulate the global compaction of chromatin.

The experiments are designed well with parameter variations necessary to test their hypotheses and suggestions from the MD simulations and are carried out well-controlled. Investigations in this topical area of chromatin biophysics are currently very active with only a few researchers doing experiments on well-controlled, highly refined, and tractable experimental systems. I believe that many researchers in this area will appreciate these timely results. Moreover, the suggestion that the effect of ISWI chromatin remodeler may be at a global level to regulate the overall chromatin compaction will be an important guide for future research. I support the publication of this manuscript provided that the following relatively minor comment is addressed in a revision.

It is unclear what the quantitative basis is to use a 300 nm wavelength for the turbidity assay. I guess this assay is based on Rayleigh scattering which should depend on the droplet size and density (or refractive index). It is unclear how the turbidity at a single wavelength of 300 nm represents the phase separation "propensity". How is this value quantitatively related to the phase separation propensity? For instance, how heavily do larger and/or denser droplets contribute to the turbidity value? Does a baseline turbidity mean no droplet formation? Or does it mean that droplet formation is limited to smaller sizes? At what time point after inducing droplet formation was the measurement made? Were the droplets matured at the time of each measurement? The authors compared the turbidity results with the simulation results side by side (figure 2) with no quantitative bases provided. Answers to these questions are important to properly evaluate the reported droplet formation "propensity" and how it can be compared to the simulation results.

(Remarks on code availability)

Reviewer #3

(Remarks to the Author)

This manuscript describes an array of theoretical and experimental techniques that seek to elucidate the effect of linker DNA on higher order structures of nucleosome arrays. The key result reveals that changes in the linker length determine the relative degree of inter- vs. intra- nucleosome interactions, effectively governing the degree of higher order organization. New results in droplet fluorescence recovery and turbidity experiments reveal systematic changes in measured condensation of nucleosome arrays. These changes are correlated to single molecule models that show reorganization of nucleosomes over the same range of linker lengths and salt concentrations. Condensation is determined by changes in the linker length, over the surprisingly short variation of 25 to 30 base pairs. Finally, remodeling proteins are shown to actively alter these lengths, thus facilitating compaction/release in vivo.

This problem has been studied, both theoretically and experimentally (hence the large number of references), and so this work will certainly be of wide interest. Theoretical and experimental studies appear carefully done and support an interesting conclusion. Comparisons over a wide range of solution conditions strengthen the correlation between the modeled structures and observed bulk results. However, it would be especially helpful to clarify why these particular modeling studies represent an improvement over previous results, many of which disagree over key details of fiber structure. Overall, this study should be published after addressing some further specific questions.

Page 4: Please provide a citation for the last sentence of the first paragraph.

Page 9 and Figures 2 and S6: Why was the diffusion normalized? This makes the numbers a bit difficult to understand. Perhaps noting the D25 value on the figures would help.

Figure 2: Comparisons between modeling and experiment are helpful as in panel c, though they would be more so with some idea of uncertainty.

Figure 2, S4 and S5: Fitted results should be summarized in a table, including error.

Figure 3: The modeling results are intriguing, especially the extension to heterogeneous linker lengths. However, it should be noted that they would be affected by dynamic release/rebinding of the outermost bases. How would nucleosome 'breathing' affect simulations?

Figure 6: Is there is some heterogeneity in the ISWI induced separations?

(Remarks on code availability)

Reviewer #4

(Remarks to the Author)

In this manuscript, Chen and colleagues address the question of how linker length affects chromatin higher-order structure using a combination of experiments and simulations. In the experiments, regular 601 arrays with different linker lengths are reconstituted into chromatin fibers and the formation of phase-separated aggregates is monitored using turbidity and FRAP essays. In parallel, coarse-grained simulations of these modeled fibers are analyzed in terms of density, diffusion coefficient, and structures. As previously reported, 10N +5 linker lengths are favorable for liquid-liquid phase separation (LLPS). The simulations largely agree with the experiments over a broad range of conditions and linker lengths. Importantly, in both

experiments and simulations, changes of 1-2 bp in linker length already had a profound effect on LLPS. Mechanistically, the simulations reveal that the internal stacking of nucleosomes within the fiber for 10N linker lengths impedes interactions with other fibers and, therefore, suppresses LLPS. These findings may significantly impact chromatin organization in vivo, and to support this, additional experiments and simulations are done using chromatin remodelers to change linker length to evaluate this effect on LLPS. The remodeling experiments confirm the essential role of linker length in organizing chromatin compaction.

The higher-order organization of chromatin is an important long-standing conundrum, and recent reports on phase separation and various structural studies have emphasized the disordered and dynamic aspects of chromatin organization, which further complicates our structural understanding. The current manuscript shows that such a structural understanding requires exquisite control or knowledge of nucleosome positioning and that variations in nucleosome positioning can significantly affect chromatin's molecular structure. The combination of experiments and simulations is well-chosen and yields new insight. Both are described in sufficient detail, and it was a pleasure to read this well-written manuscript.

I only have minor remarks:

+ Pages 7 and 8 refer to Fig. 1e, which does not exist. Should be Fig. 1d?

+ It will be good to explicitly list the sequence of the linker DNA for all experiments since nucleosome remodeling and the conformation of linker DNA may be sequence-dependent. This information is important to share for reproduction and future more detailed analysis.

+ Since the message in this paper is that LLPS is sensitively dependent on nucleosome positions, it would be good to have additional experimental conformation of these after the remodeling reactions. Fig. 6 suggests perfect five bp shifts, but the reality can be more complex and variable, especially on a DNA substrate containing 601 sequences. Did the authors attempt to verify nucleosome positions? If so, please provide additional data. If not, discuss the possibility of incomplete or heterogeneous remodeling and its possible impact on the outcomes.

(Remarks on code availability)

I am not an expert on coding, and thorough analysis of the code goes beyond my skills and time availability

Version 1:

Reviewer comments:

Reviewer #1

(Remarks to the Author)

All my concerns were addressed in the revision.

(Remarks on code availability)

The simulations are based on code that has been used and tested extensively since many years.

Reviewer #2

(Remarks to the Author)

Dear Editor,

The revision addressed my previous comments adequately. As such, I support its publication.

Sincerely,

Tae-Hee Lee

(Remarks on code availability)

Reviewer #3

(Remarks to the Author)

The authors have done a good job responding to questions, and this manuscript is certainly ready for publication!

(Remarks on code availability)

Reviewer #4

(Remarks to the Author)

My concerns have been addressed, and I can fully recommend this manuscript for publication.

(Remarks on code availability)

Version 2:

Reviewer comments:

Reviewer #1

(Remarks to the Author)

I agree with the changes and the modified way to present Fig. 6d.

(Remarks on code availability)

Reviewer #2

(Remarks to the Author)

I agree that the correction and the change made to Figure 6 do not affect the conclusion that the remodeler slides nucleosomes to modulate phase separation.

(Remarks on code availability)

Reviewer #1 (Remarks to the Author):

In this study experimentalists (Rosen lab) and computational scientists (Collepardo-Guevara lab) join forces to study liquid condensates formed from chromatin arrays. The combination of in vitro experiments and computer simulations of these two groups, who have worked on this system for several years, resulted in a manuscript that provides insight into liquid-liquid phase separation in unprecedented detail. I consider this work as a major advance in the field of chromatin research, particularly in terms of connecting small scales (nucleosomes) to larger scales (liquid condensates inside the nucleus). This is crucial because the intermediate structure, the classical chromatin fiber, still found in standard textbooks, has been understood as an artefact of in vitro experiments. We still do not have a good idea how to connect these two scales, and the current work shows how details at the small scales, in particular the DNA-linker length between nucleosomes, affect the propensity of nucleosome arrays to either stay more local or merge into a "nucleosome melt". To my knowledge, there is currently no study that shows the mechanism behind this phenomenon so clearly. I strongly recommend publishing this insightful study in Nature Communications.

We thank the reviewer for these detailed and positive comments.

The manuscript is very clearly written and I have only a few minor comments:

(1) First paragraph in the introduction: "local diffusion" of nucleosomes, please specify whether you mean in space or along the DNA molecule (I assume you mean in space).

We would prefer not to specify the kind of nucleosome diffusion as both mechanisms, nucleosome diffusion in space and nucleosome movement along the DNA fiber by remodelers, are likely relevant contributors to nucleosome dynamics observed in cells. While we agree with the referee that diffusion in space is likely the dominant contributor to motion, published data are consistent with both processes playing some role. For example, Nozaki et al. (Mol. Cell, 2017) found that ATP depletion and decreased temperature both decrease nucleosome motion. Thus, we prefer not to specify here.

¹Nozaki, T. et al. Dynamic Organization of Chromatin Domains Revealed by Super-Resolution Live-Cell Imaging. *Molecular Cell* **67**, 282-293.e7 (2017)

(2) On page 7, the authors first refer to a Fig. 1d that does not seem to quite match the actual figure and then to Fig. 1e, which seems not to exist. Please check the numbering.

We thank the reviewer for pointing out these discrepancies. We have now corrected the figure call outs.

(3) I did not fully understand the description of the computational algorithm to mimic remodeling (page 42), especially point 5. What is meant by the "sequential nucleosome neighbour", especially in the case when there are two free flanking sequences at the start.

The description of the computational algorithm has been rewritten to enhance clarity:

The algorithm chooses one fiber at random and enforces the effect of one ISWI molecule remodeling all its nucleosomes sequentially before unbinding. Remodeling is defined in our algorithm as equalizing the DNA length on both the entry and exit sides of a given nucleosome. Consequently, the algorithm starts with a solution where all chromatin arrays have the same linker DNA lengths (e.g., either all 30 bp or all 25 bp with 27 bp of DNA flanking the terminal nucleosomes on either end) and progressively produces heterogeneous mixtures of chromatin arrays with diverse distributions of linker DNA lengths. Eventually, after many iterations, the linker DNA lengths even out again, and the algorithm reaches the state of full remodeling, where all fibers have the same homogeneous distribution of DNA linker lengths (e.g., ~35 bp for the systems starting as 30-bp arrays, or ~30 bp for the systems starting as 25-bp arrays). More specifically, the algorithm defines one remodeling cycle using the following steps: (1) One chromatin fiber is chosen at random. (2) If the fiber contains flanking DNA beyond the terminal nucleosomes, the first and/or last nucleosomes are pushed to the ends of the fiber until no flanking DNAs are left. (3) One nucleosome between 2 and 11 is chosen at random. (4) The lengths of the entering and exiting linker DNAs of the chosen nucleosome are equalized, (5) The adjacent nucleosomes are remodeled next (e.g. if the 5th nucleosome was chosen in step 4, the 4th and the 6th nucleosomes are then remodeled in step 5), and step 4 is repeated until both ends of the fiber are reached.

Reviewer #2 (Remarks to the Author):

The experiments are designed well with parameter variations necessary to test their hypotheses and suggestions from the MD simulations and are carried out well-controlled. Investigations in this topical area of chromatin biophysics are currently very active with only a few researchers doing experiments on well-controlled, highly refined, and tractable experimental systems. I believe that many researchers in this area will appreciate these timely results. Moreover, the suggestion that the effect of ISWI chromatin remodeler may be at a global level to regulate the overall chromatin compaction will be an important guide for future research. I support the publication of this manuscript provided that the following relatively minor comment is addressed in a revision.

We thank the reviewer for these positive comments.

It is unclear what the quantitative basis is to use a 300 nm wavelength for the turbidity assay. I guess this assay is based on Rayleigh scattering which should depend on the droplet size and density (or refractive index). It is unclear how the turbidity at a single wavelength of 300 nm represents the phase separation "propensity".

We thank the referee for raising this issue. To clarify, we have revised the description of the assay in the Methods section on page 33, and added brief text on page 6 where the assay is first introduced. In short, as the referee intuitively, absorbance at 300 nm is used to monitor the scattering

of light by phase separated droplets with higher refractive index than the surrounding buffer and size ~300 nm or larger. By monitoring the presence of droplets as a function of salt concentration, one can identify the threshold concentration for phase separation, which can be compared between chromatin constructs.

The turbidity assay has been used by many in the field to detect the presence of phase separation^{2,3}. It has been demonstrated that this assay correlates well with other methods, such as imaging⁴ and centrifugation⁵, but offers several advantages, including unbiased interpretation, a lower sample quantity requirement, and ease of use. A variety of wavelengths have been used, and 300 nm was chosen empirically here as it gives high signal to noise. The trend between arrays remains constant at a wide range of wavelengths.

We note that we do not establish phase separation propensity from a single data point, but rather from a series of points spanning single-phase and two-phase regimes. We then compare the phase separation threshold among the different linker length constructs to give relative phase separation propensity.

²Larson, A.G. et al. Liquid droplet formation by HP1 α suggests a role for phase separation in heterochromatin. *Nature* **547**, 236-240 (2017).

³Raymond-Smiedy, P., Bucknor, B., Yang, Y., Zheng, T. & Castañeda, C.A. A Spectrophotometric Turbidity Assay to Study Liquid-Liquid Phase Separation of UBQLN2 In Vitro. in *Protein Aggregation: Methods and Protocols* (ed. Cieplak, A.S.) 515-541 (Springer US, New York, NY, 2023).

⁴Holland, J., Crabtree, M.D. & Nott, T.J. In Vitro Transition Temperature Measurement of Phase-Separating Proteins by Microscopy. in *Intrinsically Disordered Proteins: Methods and Protocols* (eds. Kragelund, B.B. & Skriver, K.) 703-714 (Springer US, New York, NY, 2020).

⁵Milkovic, N.M. & Mittag, T. Determination of Protein Phase Diagrams by Centrifugation. in *Intrinsically Disordered Proteins: Methods and Protocols* (eds. Kragelund, B.B. & Skriver, K.) 685-702 (Springer US, New York, NY, 2020).

How is this value quantitatively related to the phase separation propensity? For instance, how heavily do larger and/or denser droplets contribute to the turbidity value? Does a baseline turbidity mean no droplet formation? Or does it mean that droplet formation is limited to smaller sizes?

The reviewer raises several good points that the community is aware of but has not addressed quantitatively. The absorbance signal as a proxy for scattering is a complex convolution of the number, size, refractive index, and shape of droplets. However, when controlled for by using the same macromolecule concentration, the same incubation time, the same buffer and salt compositions, the turbidity assay has been found to yield reliable measurements of phase separation thresholds (again, by comparison to other measures such as direct observation of phase separated droplets by fluorescence microscopy), allowing for comparisons between samples.

In unpublished data, we have found that chromatin arrays do form clusters of a few molecules at baseline turbidity, as observed by dynamic light scattering and cryo-electron microscopy. However, they are transient and do not represent small, stable phase separated droplets as we do not observe macroscopic droplets under the microscope after long incubation, as would be expected from fusion of small droplets. We note that our goal in these assays is to not to characterize these small species, but rather to quantify the phase separation threshold, which we do by extrapolating backwards from conditions where droplets can be observed (see Methods pp. 33-34).

At what time point after inducing droplet formation was the measurement made? Were the droplets matured at the time of each measurement?

The measurements were consistently obtained 30 seconds after droplet induction by salt addition (now stated on p. 33). The droplets were not matured at this point. In unpublished data, we have found that absorbance at 300 nm (i.e. turbidity) decreases slowly on timescales of ~1-2 hours, likely due to droplet coalescence and settling. By collecting data for different conditions and constructs at identical timepoints, we can compare them quantitatively.

We note that the FRAP data shown in the manuscript were obtained after 2 hours of incubation, which demonstrated full recovery. Furthermore, these droplets remain liquid after 24 hours, with similar fluorescence recovery, sphericity, and increase in size consistent with continued fusion. So droplets do not appear to be changing over time.

The authors compared the results with the simulation results side by side (figure 2) with no quantitative bases provided. Answers to these questions are important to properly evaluate the reported droplet formation "propensity" and how it can be compared to the simulation results.

We hope that our answers above clarify use of the turbidity assay to compare the phase separation thresholds of different constructs, and thus their relative propensities to phase separate. As detailed on pp. 41-2 of the Methods section, the simulations determine the phase separation threshold by quantifying the density of molecules in the dense and dilute phases of simulations performed at different salt concentrations. This allows the coexistence curve to be mapped and the threshold salt concentration to be determined.

Reviewer #3 (Remarks to the Author):

This problem has been studied, both theoretically and experimentally (hence the large number of references), and so this work will certainly be of wide interest. Theoretical and experimental studies appear carefully done and support an interesting conclusion. Comparisons over a wide range of solution conditions strengthen the correlation between the modeled structures and observed bulk results. However, it would be especially helpful to clarify why these particular modeling studies represent an improvement over previous results, many of which disagree over key details of fiber structure. Overall, this study should be published after addressing some

further specific questions.

Page 4: Please provide a citation for the last sentence of the first paragraph.

We have added a citation to the sentence “Studies of these systems also revealed a periodic relationship between the length of internucleosomal linker DNA and LLPS propensity”

Page 9 and Figures 2 and S6: Why was the diffusion normalized? This makes the numbers a bit difficult to understand. Perhaps noting the D25 value on the figures would help.

The diffusion coefficient calculated from the simulations is not physically meaningful because simulation time does not correspond to real time. Hence, we display a normalized diffusion coefficient relative to 25 bp arrays to demonstrate relative change.

Figure 2, S4 and S5: Fitted results should be summarized in a table, including error.

Supplemental Table 2 now has summaries of fitted results regarding Figure 1, 2, and Extended Data Figure 4, 5.

Figure 3: The modeling results are intriguing, especially the extension to heterogenous linker lengths. However, it should be noted that they would be affected by dynamic release/rebinding of the outermost bases. How would nucleosome ‘breathing’ affect simulations?

We agree with the reviewer that nucleosome breathing is an important feature of nucleosomes and chromatin. The simulations do include nucleosome breathing, and in fact we previously found that breathing is an important contributor to chromatin phase separation⁶.

⁶Farr, S.E., Woods, E.J., Joseph, J.A., Garaizar, A. & Collepardo-Guevara, R. Nucleosome plasticity is a critical element of chromatin liquid–liquid phase separation and multivalent nucleosome interactions. *Nature Communications* **12**, 2883 (2021).

Figure 6: Is there is some heterogeneity in the ISWI induced separations?

Fig. 6 and Extended Data Fig. 10 now have additional data on the ISWI experiments, which reveal an increase in the heterogeneity of linker lengths during active remodeling, as well as an increase in linker length consistent with our design. We have added discussion of these new data on p. 16.

Reviewer #4 (Remarks to the Author):

The higher-order organization of chromatin is an important long-standing conundrum, and recent reports on phase separation and various structural studies have emphasized the disordered and

dynamic aspects of chromatin organization, which further complicates our structural understanding. The current manuscript shows that such a structural understanding requires exquisite control or knowledge of nucleosome positioning and that variations in nucleosome positioning can significantly affect chromatin's molecular structure. The combination of experiments and simulations is well-chosen and yields new insight. Both are described in sufficient detail, and it was a pleasure to read this well-written manuscript.

We thank the reviewer for his/her thorough evaluation of our study and positive comments regarding its potential impact.

I only have minor remarks:

+ Pages 7 and 8 refer to Fig. 1e, which does not exist. Should be Fig. 1d?

We thank the reviewer for pointing out these discrepancies. We have now corrected the figure call outs.

+ It will be good to explicitly list the sequence of the linker DNA for all experiments since nucleosome remodeling and the conformation of linker DNA may be sequence-dependent. This information is important to share for reproduction and future more detailed analysis.

We agree with the reviewer that the linker DNA sequence is an important consideration for this study. We have now added Supplemental Table 1 with the full sequences of all arrays in our study. We also plan to deposit plasmids containing these arrays to Addgene to enable their use by the community.

+ Since the message in this paper is that LLPS is sensitively dependent on nucleosome positions, it would be good to have additional experimental conformation of these after the remodeling reactions. Fig. 6 suggests perfect five bp shifts, but the reality can be more complex and variable, especially on a DNA substrate containing 601 sequences. Did the authors attempt to verify nucleosome positions? If so, please provide additional data. If not, discuss the possibility of incomplete or heterogeneous remodeling and its possible impact on the outcomes.

We thank the reviewer for this comment, and now provide additional data in Fig. 6 and Extended Data Fig. 10 to demonstrate that the remodeling is incomplete and heterogeneous in our reactions, but towards the direction that we designed. The new data provide additional insight to the sensitivity of the array towards condensation. Discussion of these data is included on p. 16.

Reviewer #4 (Remarks on code availability):

I am not an expert on coding, and thorough analysis of the code goes beyond my skills and time availability.